# A modular platform for automated cryo-FIB workflows

Sven Klumpe[1†], Herman KH Fung[2†], Sara K Goetz[2,3†], Ievgeniia Zagoriy[2], Bernhard Hampoelz[2‡], Xiaojie Zhang[2], Philipp S Erdmann[1§], Janina Baumbach[2‡], Christoph W Müller[2], Martin Beck[2,4‡], Jürgen M Plitzko[1*], Julia Mahamid[2*]

[1]Department Molecular Structural Biology, Max Planck Institute of Biochemistry, Martinsried, Germany; [2]Structural and Computational Biology Unit, European Molecular Biology Laboratory, Heidelberg, Germany; [3]Collaboration for joint PhD degree between EMBL and Heidelberg University, Faculty of Biosciences, Heidelberg, Germany; [4]Cell Biology and Biophysics Unit, European Molecular Biology Laboratory, Heidelberg, Germany

**\*For correspondence:**
plitzko@biochem.mpg.de (JMP);
julia.mahamid@embl.de (JM)

[†]These authors contributed equally to this work

**Present address:** [‡]Department Molecular Sociology, Max Planck Institute of Biophysics, Frankfurt, Germany; [§]Fondazione Human Technopole, Milan, Italy

**Abstract** Lamella micromachining by focused ion beam milling at cryogenic temperature (cryo-FIB) has matured into a preparation method widely used for cellular cryo-electron tomography. Due to the limited ablation rates of low $Ga^+$ ion beam currents required to maintain the structural integrity of vitreous specimens, common preparation protocols are time-consuming and labor intensive. The improved stability of new-generation cryo-FIB instruments now enables automated operations. Here, we present an open-source software tool, SerialFIB, for creating automated and customizable cryo-FIB preparation protocols. The software encompasses a graphical user interface for easy execution of routine lamellae preparations, a scripting module compatible with available Python packages, and interfaces with three-dimensional correlative light and electron microscopy (CLEM) tools. SerialFIB enables the streamlining of advanced cryo-FIB protocols such as multi-modal imaging, CLEM-guided lamella preparation and in situ lamella lift-out procedures. Our software therefore provides a foundation for further development of advanced cryogenic imaging and sample preparation protocols.

## Editor's evaluation

Since its initial inception, as a sample thinning technique for cryo-electron tomography, cryo-focused ion beam (FIB) milling has developed to include a range of different methodologies. At the moment, there is no dedicated software that is able to integrate all these methodologies and their supporting software packages. Klumpe et al. aimed to alleviate this problem by developing an open-source software tool, SerialFIB. SerialFIB allows users to set up automated protocols for on-grid lamella preparation, FIB-SEM volume imaging, and lift-out trench milling. This work has significant importance for the field as it decouples the need for proprietary software for the execution of highly specialized milling protocols.

## Introduction

Cryo-electron tomography (cryo-ET) is a structural biology technique that can reveal the sociology of macromolecules in their native environment (*Beck and Baumeister, 2016*). Fixation through rapid cooling to cryogenic temperature (below –140°C; *Dubochet and McDowall, 1981*; *Glaeser and Taylor, 1978*) arrests water molecules in a glass-like phase, thus preserving biological structures in a near-native state. The vitrification temperature is pressure dependent. Hence, the maximum achievable

vitrification depth ranges from ~10 μm for plunge-freezing at ambient pressure to ~200 μm in high-pressure freezing (HPF) (*Dubochet, 1995*). Most cellular cryo-ET imaging is performed in a transmission electron microscopy (TEM) at 300 kV, at which the inelastic mean free path of an electron in biological material is in the order of 300–400 nm. Consequently, thicker samples will exhibit a rapidly decreasing signal-to-noise ratio (*Yonekura et al., 2006*; *Rice et al., 2018*; *Diebolder et al., 2012*). Therefore, most cells, excluding the thinnest peripheries of adherent eukaryotic cells, are not directly suitable for cryo-ET. Cryo-FIB micromachining enables the production of electron-transparent cellular slices thinner than 300 nm, which are termed lamellae (*Marko et al., 2007*; *Rigort et al., 2012*). Cryo-FIB milling has been successfully applied to single-cell specimens deposited or grown on TEM grids, or to voluminous multicellular samples from HPF using in situ lift-out (*Mahamid et al., 2015*; *Schaffer et al., 2019*). These approaches yield specimens of suitable quality for molecular-resolution cryo-ET (*Pfeffer and Mahamid, 2018*), circumventing artifacts described for mechanical sectioning by cryo-ultramicrotomy (*Al-Amoudi et al., 2005*). However, common cryo-FIB preparations constitute a low-throughput method owing to slow ablation rates of $Ga^+$ ions at low currents (30 pA to 1 nA) typically used to micromachine vitreous specimens. The method further requires significant user expertise with a steep learning curve, which limits its practical usability. In addition, emerging concepts such as multi-modal imaging through serial cryo-FIB ablation and scanning electron microscopy (SEM) volume imaging coupled to lamella preparation are challenging to achieve with manual operations (*Wu et al., 2020*; *Zhu et al., 2021*; *Spehner et al., 2020*). Automation is thus essential to improving the performance, throughput, and applicability of cryo-FIB methods as has been the case for cryo-TEM imaging (*Mastronarde, 2018*; *Carragher et al., 2000*; *Nickell et al., 2005*).

Automation for cryo-FIB , designed for the specialized task of routine on-grid lamellae preparation from single-cell specimens, has been to date based on proprietary software (*Zachs et al., 2020*; *Tacke et al., 2021*; *Kuba et al., 2021*) or command-line operation (*Buckley et al., 2020*). An open-source and easy-to-use package for customized cryo-FIB-SEM workflows is currently missing. Inspired by SerialEM (*Mastronarde, 2005*), we developed an automation software platform to address the current gap for cryo-FIB instruments. We demonstrate the flexibility of this software platform on several use cases ranging from standard preparations to customized applications. This includes (i) on-grid lamella preparation, (ii) cryo-fluorescence light microscopy (cryo-FLM)-based 3D-targeted lamella preparation, (iii) cryo-FIB-SEM volume imaging, with an option for subsequent lamella preparation, and (iv) micromachining for in situ lift-out from voluminous HPF samples. Apart from the proprietary application programming interface (API) required for communication with the microscope, the software code is open source, opening the potential for further workflow development. We show that our milling protocols can be performed without supervision on six different specimens. Some of the specimens contain high-density intracellular structures, that is, lipid droplets or minerals, which introduce additional difficulty to the generation of homogenously thin lamellae. We therefore provide milling protocols for a broad range of cellular sample types. We demonstrate that cryo-FIB procedures such as cryo-FIB-SEM volume imaging and lift-out trench milling can be user-tailored in SerialFIB. Thus, automation can increase the throughput for advanced imaging and sample preparation workflows as time-consuming steps can be performed overnight provided compatibility of the instrument for prolonged cryogenic operation times.

## Results

### Software design

In developing SerialFIB, we seek to provide a flexible, Python-based platform for the creation and execution of customized cryo-FIB milling protocols (*Figure 1*). The graphical user interface (GUI) and image processing steps are therefore decoupled from underlying communications with the cryo-FIB-SEM instrument. Communication with the instrument is handled through an intermediary 'driver,' which makes use of the microscope's API to perform beam, stage, and imaging operations (*Figure 1*). Here, we developed a driver for Thermo Fisher Scientific systems based on the proprietary API AutoScript 4. Through this driver, we tested the execution of milling protocols using SerialFIB on a Scios and two Aquilos systems. So long as an API is provided by the microscope manufacturer, new drivers for other cryo-FIB systems can be developed. The feasibility and requirements for extending SerialFIB to presently available systems are detailed in Appendix 1.

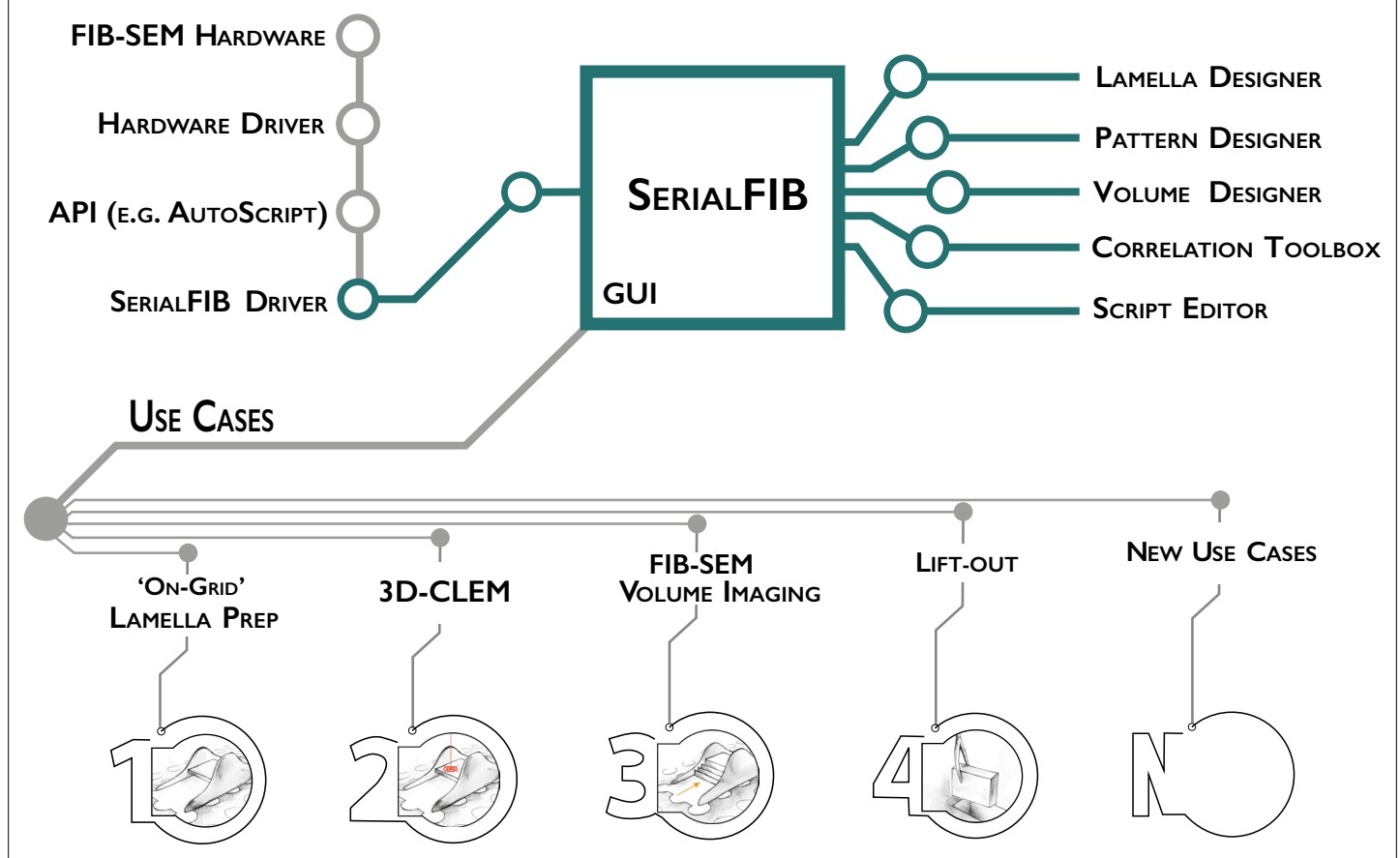

**Figure 1.** Software architecture, modules, and use cases of SerialFIB. Developments presented in this work are highlighted in green. The graphical user interface (GUI) is largely decoupled from instrument operations, which are controlled by the developed SerialFIB driver. Modules (right) enable design of protocols for different use cases (bottom, 1–4), and a scripting interface that offers flexibility for new developments (bottom, N).

The online version of this article includes the following figure supplement(s) for figure 1:

**Figure supplement 1.** Graphical user interface (GUI) for automated cryo-focused ion beam (cryo-FIB) protocols.

**Figure supplement 2.** SerialFIB interface for volume imaging and Script Editor.

The basic unit for milling or patterning tasks in SerialFIB are 'pattern sequence files', wherein a sequence of milling operations characterized by the milling current, milling time, pattern size, and pattern position relative to a user-defined reference point are defined. We also provide 'protocols', text-based files that are translated by SerialFIB into pattern sequence files. Protocols simplify the creation of pattern sequence files by focusing on use-case-specific parameters. For example, for on-grid lamella preparation, the parameters specified are the number of milling steps, the current and milling time in each step, and the upper to lower milling pattern distance. Currently, we provide protocols for several use cases including on-grid lamella preparation, 3D cryo-FLM-guided lamella preparation, and FIB-SEM volume imaging. More customized tasks employ pattern sequence files directly, as has been implemented here for automated lift-out site preparation. Upon selection of a protocol in the SerialFIB GUI (*Figure 1—figure supplement 1A*), the software automatically generates the necessary pattern sequence files, which are then parsed and executed through the system-specific intermediary driver. Protocols and pattern sequence files can be edited in a text editor or through one of the dedicated 'Designer' modules, which provides a graphical interface and preview of parameters to enable adjustments during optimization of the workflow (*Figure 1—figure supplement 1B and C*).

During a typical cryo-FIB session, points of interest on the sample and their coincidence height (the height at which the FIB and SEM beam coincide) are identified manually through the microscope's user interface. Then, in SerialFIB, positions are stored, reference images for realignment are acquired, and the target position and milling geometry are defined on the reference image per position of

interest. For on-grid lamella preparation, for example, this means defining the lamella position, width, and extreme points above and below the target lamella from which ablation of material will begin (*Figure 1—figure supplement 1A*). To start the procedure, the corresponding protocol is initiated. This will trigger the creation and execution of pattern sequence files for each patterning step as defined in the protocol. The setup within the SerialFIB GUI is standardized, allowing for the execution of several protocols at the same position, for example, FIB-SEM volume imaging (*Figure 1—figure supplement 2A*) followed by lamella preparation (*Figure 1—figure supplement 1B*).

FIB milling routines that are not covered by our predefined protocols, for example, Waffle method (*Kelley et al., 2020*), can be developed through customized pattern sequence files and the 'ScriptEditor' module (*Figure 1—figure supplement 2B*). In the ScriptEditor module, images and stage positions previously defined in the GUI are accessible together with the underlying Python commands of the driver. Thus, additional use cases based on custom Python scripts can be implemented in SerialFIB and shared. Examples for scripts are given on the GitHub repository.

## Automated on-grid lamellae preparation

Similar to recent developments (*Zachs et al., 2020*; *Tacke et al., 2021*; *Buckley et al., 2020*), SerialFIB offers an automated solution for preparing on-grid lamellae for cryo-ET. For each position of interest, the user defines the target lamella width and position, and extreme milling points on the reference image (*Figure 1—figure supplement 1A*). The definition of extreme milling points ensures that the milling of grid bars and other objects, which can deflect the ion beam and cause damage to the lamella, is avoided (*Figure 2A*). Then, the user can choose to create micro-expansion joints (*Wolff et al., 2019*) to relieve tension in the frozen specimen (*Figure 2B*). This is followed by removal of material in two stages: (i) rough milling, where all positions of interest are milled to ~1 µm thickness, and (ii) fine milling, where the generated slices are thinned to ~200 nm thickness at the end of the session to achieve electron transparency (*Figure 2C*). Amorphous ice condensation on the final lamellae due to residual water in the microscope, which reduces image contrast, can be minimized by limiting the total fine milling time for each grid (*Schaffer et al., 2017*). A condensation rate of 50 nm/hr is reported in the specifications of our systems. Thus, balancing between the number of lamellae generated and loss of contrast due to condensation, we limited our fine milling time to 1 hr.

Automated milling in SerialFIB uses a series of realignments to precisely recall the target position for each milling step (*Figure 2—figure supplement 1*). The realignment procedure is based on image cross-correlation (*Guizar-Sicairos et al., 2008*). It is performed when navigating between stored stage positions and when changing between FIB currents to compensate for imperfect microscope alignments. In the first case, stage movements are used for large distance offsets (>10 µm) and beam shifts are applied for subtle corrections. A low ion beam current (~10 pA) is used to capture the current view after recalling a stored position for registration with the stored reference image. When changing between FIB currents, two images are taken, one at the previous beam current to serve as a reference, and one at the new beam current to determine the offset. Here, only beam shifts are used for realignment. Currently, we correct for stage drift when calling a stored stage position, but not between milling steps as we observed lower stage drift rates than the specifications of the instrument provider (60 nm/min).

To benchmark the automated workflow, we applied on-grid lamella milling to five different cell types with adjusted protocols that are provided on our GitHub repository (*Figure 2*, *Figure 2—figure supplements 2–3*, *Supplementary file 1*, 'Data availability' section). For all samples, lamellae with thickness of roughly 1 µm were generated in three steps with FIB currents gradually decreasing from 1 to 0.3 nA as in manual operation (*Rigort et al., 2012*; *Schaffer et al., 2015*). After completion of rough milling for all positions, fine milling to the desired lamella thickness was performed in one or multiple steps with decreasing FIB currents. For *Emiliania huxleyi*, *Chlamydomonas reinhardtii*, and Sum159 cells, we found that multiple steps with decreasing FIB currents during fine milling helped to reduce curtaining artifacts caused by mineral crystals or high lipid droplet content (*Supplementary file 1*). This underscores the need to adjust milling steps and parameters empirically when optimizing a sample towards on-grid lamella preparation.

Out of a total of 77 positions, we generated 64 lamellae that were 5.6–25.6 µm in width as determined by SEM and 70–470 nm in thickness as determined by cryo-ET, providing an overall milling success rate of 83% (*Table 1*, *Figure 2—figure supplement 4*).

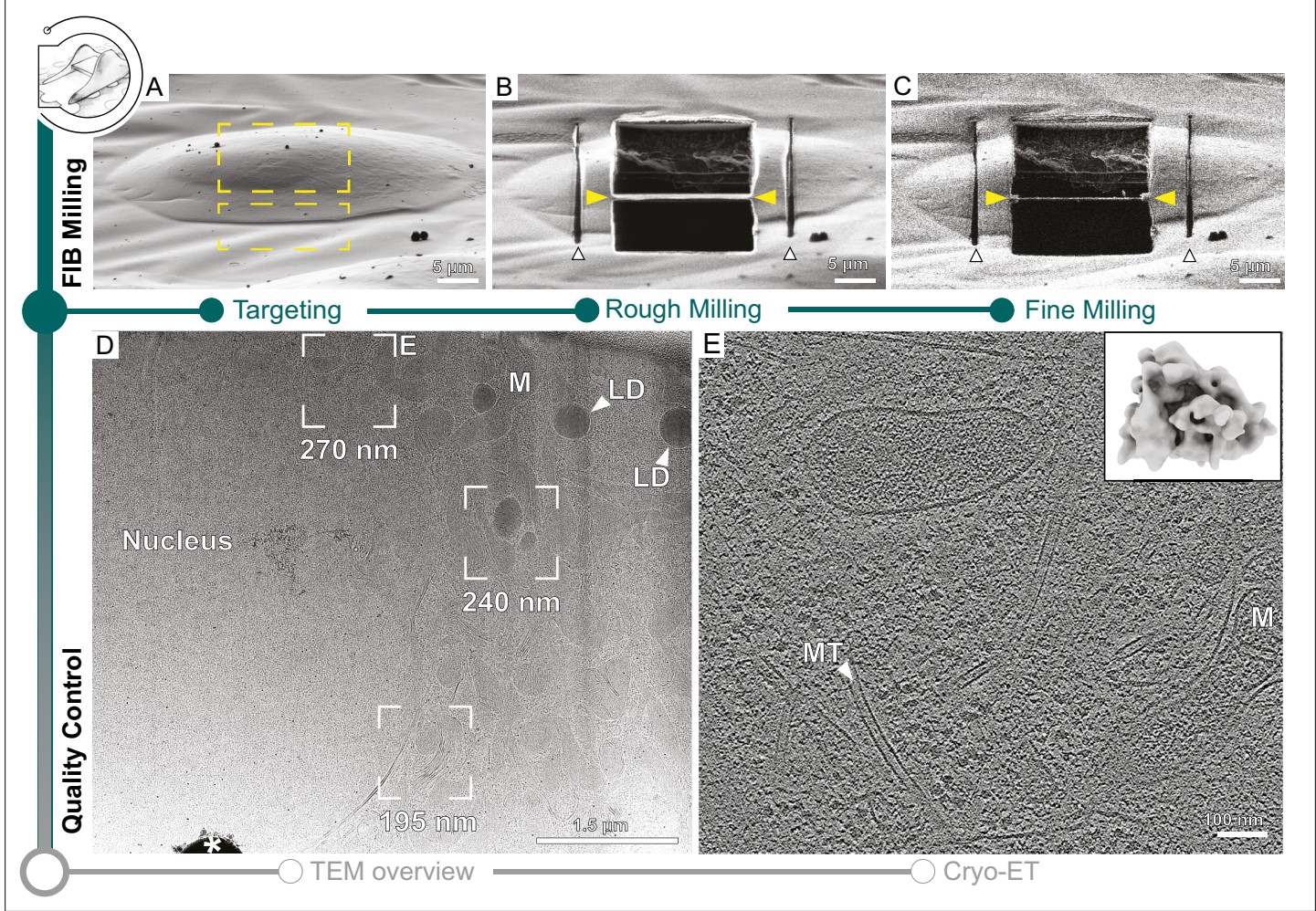

**Figure 2.** Automated on-grid lamella preparation of Sum159 breast cancer cells. (**A**) Focused ion beam (FIB) image of a cell prior to lamella preparation. Yellow rectangles indicate milling patterns where material is subsequently removed. (**B**) Micro-expansion joints (white arrowheads) and lamella (yellow arrowheads) milled to a target thickness of 1 μm. (**C**) Lamella after fine milling to a target thickness of 300 nm. (**D**) Transmission electron microscopy (TEM) overview of lamella in (**C**). Frames indicate examples of tilt series acquisition positions (out of eight acquired on this lamella) and the local thickness determined from reconstructed tomograms. Nucleus, lipid droplets (LD), and mitochondria (M) are observable. * indicates an ice crystal introduced during transfer between FIB and TEM. Area indicated by (**E**) is enlarged. (**E**) A slice through the tomogram depicts the cytosol with microtubules (MT) and a mitochondrion (M). Inset shows a ribosome subtomogram average determined from the dataset collected on this single lamella (four tomograms; 4378 subtomograms; 24 Å resolution).

The online version of this article includes the following video and figure supplement(s) for figure 2:

**Figure supplement 1.** Functions executed by SerialFIB for preparation of on-grid lamellae.

**Figure supplement 2.** Automated on-grid lamella preparation of HeLa cells.

**Figure supplement 3.** Automated on-grid lamella preparation of *E. huxleyi* cells.

**Figure supplement 4.** Width (**A**) and thickness (**B**) of successfully prepared lamellae from five eukaryotic cell types.

**Figure 2—video 1.** Tomographic volume of a Sum159 breast cancer cell related to Figure 2E depicting the cytosol with microtubules (MT) and a mitochondrion (Mito) next to the nucleus.

https://elifesciences.org/articles/70506/figures#fig2video1

**Figure 2—video 2.** Tomographic volume of a HeLa cell related to Figure 2—figure supplement 2E depicting the nuclear periphery, and the cytosol with microtubules (MT) and a lipid droplet (LD).

https://elifesciences.org/articles/70506/figures#fig2video2

**Figure 2—video 3.** Tomographic volume of *E. huxleyi* cells related to Figure 2—figure supplement 3E depicting the cytosol with the basal body of a cilium.

https://elifesciences.org/articles/70506/figures#fig2video3

**Table 1.** Statistics and success rates of automated on-grid lamellae milling with SerialFIB for five different cell types.

| Sample | # Target sites | # Fine-milled lamellae | # Successfully transferred to TEM | Lamella thickness (nm) |
|---|---|---|---|---|
| Sum159 | 22 | 19 | 17 | 70–410 |
| HeLa | 22 | 18 | 18 | 100–450 |
| *E. huxleyi* | 9 | 9 | 9 | 175–470 |
| *C. reinhardtii* | 16 | 10 | n/a | 140–350 |
| *S. cerevisiae* | 8 | 8 | 8 | 190–300 |
| Total | 77 | 64 | | |
| Success rate (%) | | 83.1 | | |

TEM, transmission electron microscope.

## 3D-CLEM-targeted lamella preparation

When studying rare biological events, CLEM approaches are indispensable for cryo-FIB-milling as the final tomogram volume captures only a tiny fraction of the entire cell (~0.01 and ~0.1% for HeLa and *C. reinhardtii*, respectively) (*Wu et al., 2020*; *Schorb et al., 2017*; *Plitzko et al., 2009*). Using fiducials, fluorescent microbeads for example, which are visible in fluorescence, SEM and FIB imaging, cryo-FLM data can be superposed with SEM and FIB images to localize a structure of interest for targeted lamella preparation. Currently, resolution in cryo-FLM imaging is limited due to the use of dry objectives. For the confocal laser scanning system employed in this work, which has an objective of numerical aperture 0.9, lateral resolution is limited to 215–442 nm and axial resolution to 933–1915 nm at a pinhole size of 1 Airy unit across the visible spectrum. The previously described 3D Correlation Toolbox (3DCT) provides an interface to calculate the geometric transform between imaging modalities (*Arnold et al., 2016*). To improve the precision of localization beyond the resolution limit, Gaussian fitting is performed for the determination of fiducial centers in the fluorescence modality. To further improve on the throughput and usability of this technically challenging workflow, we have introduced new features in 3DCT to assist with fiducial selection and superposition of the cryo-FLM data (*Figure 3—figure supplement 1*). Using HeLa cells stained live with mitochondria and lipid droplet small-molecule dyes, we demonstrate here the new features and illustrate how SerialFIB can import correlated coordinates from 3DCT for automated lamella preparation (*Figure 3*).

The presence of ice contaminants and the shallow angle of the ion beam view make identification of the microbead fiducials in a FIB image difficult. Thus, correlation is often performed first with the SEM image. To aid in this process, we implemented a bead detection tool based on cross-correlation with a Gaussian kernel and Hough transform to facilitate semi-automated detection of beads in the SEM image (*Figure 3—figure supplement 1A*), after which corresponding beads in the cryo-FLM data can be selected in the 3DCT interface for calculation of a coordinate transform (*Arnold et al., 2016*). For fast validation of the calculated transform, maximum intensity projections of the 3D cryo-FLM data can now be generated within 3DCT and overlaid onto the SEM image (*Figure 3A*). Once the positions of fiducials have been established, their corresponding positions in the FIB image need to be determined. The matching of fiducials between SEM and FIB images was previously performed manually. The new 'Fiducial Rotation' feature in 3DCT (*Figure 3—figure supplement 1B*) takes advantage of the fact that at coincidence height SEM and FIB images are related by the angle between the two beams and beam shifts. By rotating and translating fiducial coordinates with respect to a user-defined reference point in both images (*Figure 3—figure supplement 1B*), for example, surface ice contaminants or similarly recognizable feature, the positions of fiducials in the FIB image are determined. These positions can be adjusted subsequently within the 3DCT GUI in an iterative manner to minimize registration residuals and a final transform calculated. For automated milling, the correlated FIB image is loaded into SerialFIB to serve as the reference image with marked target position of structures of interest. Maximum intensity projections can be generated and overlaid onto the displayed FIB image to help define milling positions (*Figure 3B*). Additionally, transformed coordinates of points of interest

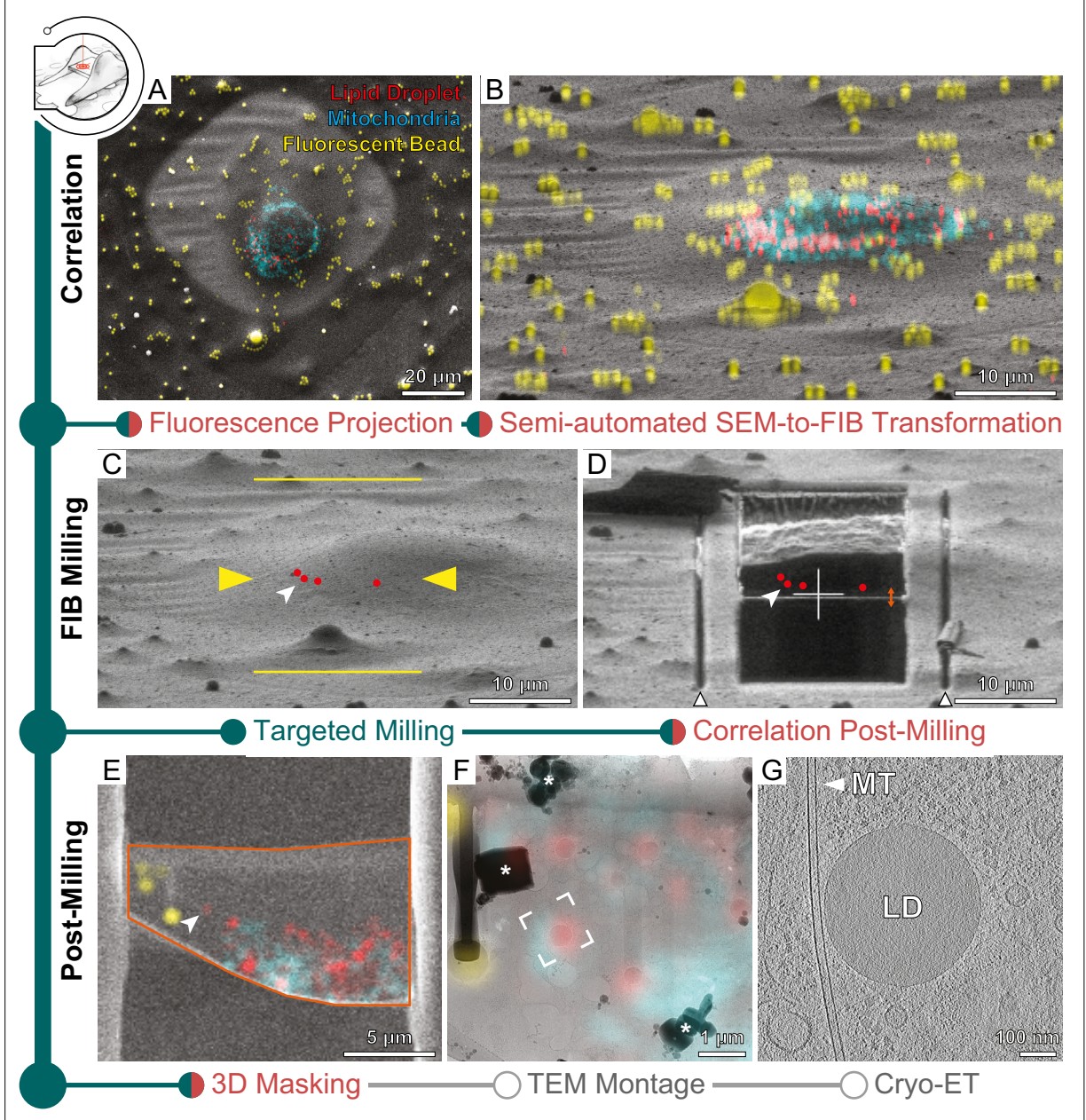

**Figure 3.** Three-dimensional correlative light and electron microscopy (3D-CLEM)-targeted lamella preparation of HeLa cells. Green modules represent tasks performed with SerialFIB; orange, with 3D Correlation Toolbox (3DCT). (**A**) Scanning electron microscopy (SEM) and (**B**) focused ion beam (FIB) images of a cell. Overlaid are maximum intensity projections of cryo-fluorescence light microscopy (cryo-FLM) signal corresponding to lipid droplets (red), mitochondria (cyan), and beads (yellow). Correlation with the FIB image is aided by 3D transformation of fiducials detected in the SEM image. (**C**) Correlated positions of lipid droplets (red dots) imported into SerialFIB. White arrowhead indicates the targeted lipid droplet. Yellow arrowheads indicate the targeted lamella position; yellow lines indicate milling extreme positions. (**D**) Final lamella after automated FIB milling. Red dots indicate the position of lipid droplets following re-correlation with the FIB image after milling. White triangles highlight micro-expansion joints. (**E**) Masking of FLM volumes based on lamella outline in the SEM image (orange) and lamella position in the FIB image (orange double arrow in **D**). A 300 nm FLM virtual slice centered on the lamella is shown. (**F**) Overlay of the best-fitting fluorescence plane with the transmission electron microscopy (TEM) image of the lamella. Different heights were sampled in relation to the FIB image post-milling to determine the plane where bead and lipid droplet fluorescence signal overlaps best with the TEM image (see Materials and methods). White frame indicates the targeted lipid droplet and area of cryo-ET acquisition. * denotes ice crystal contamination from transfers. (**G**) Tomographic slice of area indicated in (**F**) depicts the cytosol with a microtubule (MT) and a lipid droplet (LD).

The online version of this article includes the following video and figure supplement(s) for figure 3:

**Figure supplement 1.** New 3D Correlation Toolbox (3DCT) features.

*Figure 3 continued on next page*

*Figure 3 continued*

**Figure supplement 2.** Examples of successful 3D-targeted milling of lipid droplets.

**Figure supplement 3.** Local deformations of specimens.

**Figure 3—video 1.** Tomographic volume of a HeLa cell related to Figure 3G depicting the cytosol with microtubules (MT), a lipid droplet (LD), and a multivesicular body (MVB).

https://elifesciences.org/articles/70506/figures#fig3video1

---

calculated in 3DCT can be imported directly into the SerialFIB GUI (*Figure 3C*). Automated milling as described above can then be executed to generate lamellae that retain the structure of interest (*Figure 3D*).

Finally, a new 3DCT feature offers the possibility to overlay the cryo-FLM data with the SEM images of the generated lamella for subsequent navigation during cryo-ET data acquisition (*Figure 3E–G*). A 3D mask describing the thickness, orientation, and boundary of the lamella can be created and applied to the original fluorescence volume before projection onto the SEM image (*Figure 3—figure supplement 1C*). We term the resulting masked fluorescence volume a FLM 'virtual slice', which provides a way to visualize signal contained within the lamella (*Figure 3E*). For guiding cryo-ET data acquisition, we make use of the 2D affine image registration feature of SerialEM to relate between the SEM/fluorescence overlay and TEM montage of the lamella (*Figure 3F*). Thereafter, positions are defined for automated tilt-series collection (*Figure 3F and G*).

To evaluate the determinants of successful 3D-CLEM-guided lamella preparation, we targeted lipid droplets in 14 HeLa cells over four separate sessions (*Figure 3*, *Supplementary file 2*). Targeted lipid droplets measured less than 500 nm in diameter, which is comparable to the resolution of the cryo-FLM system used, and thus make a good case study for evaluating this workflow. Cells were plunge-frozen on two different substrates: titanium grids with holey 1/20 $SiO_2$ support (1 µm holes positioned 20 µm apart), and the more widely used gold grids with holey 1/4 $SiO_2$ support. By comparing between the final TEM image and FLM virtual slice based on correlations with the pre-milling FIB image, we found that the targeted lipid droplet was retained in 7 out of 10 lamellae on titanium $SiO_2$ 1/20 across three grids processed in separate sessions. In contrast, the targeted lipid droplet was retained in zero out of four lamellae on gold $SiO_2$ 1/4 from one grid (*Supplementary file 2*, *Figure 3—figure supplement 2*). However, upon comparison between the final TEM image and FLM virtual slice based on FIB images acquired after milling, none of the target positions coincided with the lamella any more (*Supplementary file 2*). To investigate this mismatch, we generated virtual slices corresponding to different sample heights in the FIB image and projected these slices onto the TEM image. By assessing qualitatively how well bead and lipid droplet fluorescence in the different virtual slices corresponded to features on the TEM image, we determined which sample height in the FIB image generated the best-fitting virtual slice (*Figure 3—figure supplement 2G–I*). We found that the best-fitting slice was within 3 pixels in the FIB y-direction (corresponding to 253–506 nm) of the correlated target position post-milling for 9 out of 10 lamellae on titanium $SiO_2$ 1/20, and more than 10 pixels away (>670 nm) for all lamellae on gold $SiO_2$ 1/4 (*Supplementary file 2*, *Figure 3—figure supplement 2*). These results hint at the possibility of local deformations within the sample occurring during FIB milling, which can impinge on the success rate of 3D-CLEM-guided workflows beyond inaccuracies in milling and correlation. Indeed, elastic 2D B-spline registration of FIB images before and after milling revealed a nonuniform displacement in the support of the grid squares (*Figure 3—figure supplement 3*). This displacement occurred irrespective of whether rough and fine milling were carried out separately or in one continuous sequence per position. In all cases, regardless of support type, the displacements seemed more pronounced in areas of thin ice and along the y-direction of the FIB image (up to 2.5 µm). Despite this, it was still possible to retain the target lipid droplet during milling in 70% of the tested cases on titanium support. Taken together, our observations suggest that mechanical stability of the specimen should be considered when performing 3D-CLEM-guided milling for retaining structures of interest in the final lamella. It appears that the combination of a rigid titanium grid, a more continuous support (1/20), and similar thermal expansion behavior between grid and support can help to mitigate local deformations around the milling site. However, a more systematic evaluation would be needed to confirm these observations. In cases of mechanical deformations that compromise correlation accuracy, intracellular fiducials, for example, by lipid droplet staining

(*Scher et al., 2021*; *Okolo et al., 2021*; *Klein et al., 2021*), can be leveraged to provide local reference points for refinement of the correlation post-milling across modalities.

## Cryo-FIB-SEM volume imaging

Serial cryo-FIB sectioning and SEM imaging produce a nanometer-scale 3D representation of the specimen, providing valuable cellular context on organelle size and distribution (*Wu et al., 2020*; *Zhu et al., 2021*; *Scher et al., 2021*; *Okolo et al., 2021*; *Klein et al., 2021*) and an opportunity to fine-tune the positioning of lamella preparation. Once a feature of interest is identifiable in the SEM images, volume imaging can be stopped to allow subsequent lamella generation. With SerialFIB, we provide an automated cryo-FIB-SEM volume imaging workflow. We tested this workflow on Sum159 breast cancer cells. First, an opening into the cell is created by FIB ablation at the cell's uppermost edge. Then, a volume to be imaged is chosen, within which cellular material is alternatingly ablated in defined steps and the newly exposed surface is imaged by the SEM (*Figure 4A–E*, *Figure 4—video 1*). SEM imaging conditions are optimized to visualize cellular structures, while keeping acquisition times as short as possible (*Figure 4—figure supplement 1A*, see Materials and methods). As line integration and higher cumulative dwell times are advantageous for noise reduction in cryo-SEM imaging of unstained samples (*Spehner et al., 2020*), the choice of imaging parameters is a trade-off between acquisition time, image quality, and applied dose (between 0.5 and 1.5 e/$\text{Å}^2$ per slice; *Goggin et al., 2020*). For *C. reinhardtii* cells as an example, improvement of the signal-to-noise ratio via line integration, with the drawback of longer acquisition times, enabled resolving nuclear pore complexes, the nucleolus, and the Golgi apparatus (*Figure 4—figure supplement 1E–G*).

After volume imaging, the stage positions and patterns within the GUI employed for SEM volume acquisition can be reused to perform lamella preparation (*Figure 1—figure supplement 1*, stage positions and patterns). Alternatively, the position of lamella generation can be refined manually in the GUI (*Figure 1—figure supplement 1*, *Figure 4B and C*). In any case, 50–100 nm should be ablated from the last SEM imaging surface to ensure that material potentially damaged by previous imaging and milling is removed. Tomograms acquired from such lamellae are visually indistinguishable from those generated by conventional on-grid preparations (*Figure 4F*, *Figure 4—video 2*). Ribosome averages from lamellae generated without prior volume imaging reached 24 Å resolution from 4378 subtomograms (*Figure 2E* inset), while averages from lamellae that were previously volume-imaged reached 24 Å resolution from 3380 subtomograms (*Figure 4F* inset), indicating that both approaches produce lamellae of similar quality. Streaks in the raw FIB-SEM volume data (*Figure 4D*, *Figure 4—figure supplement 1A*) resulting from curtaining by dense objects such as lipid droplets can be corrected for using wavelet decomposition (*Figure 4—figure supplement 1B*; *Spehner et al., 2020*). A mask produced by Gaussian blurring and image erosion can compensate for charging artifacts (*Figure 4—figure supplement 1C*). Subsequent local contrast enhancement can restore cellular details (*Figure 4E*, *Figure 4—figure supplement 1D*). We provide a script for the postprocessing of SEM images (see 'Data availability') that allowed better identification of cellular structures (*Figure 4—figure supplement 1E–G*) throughout the volume, exemplified by lipid droplets and the nucleus (*Figure 4E* inlay, *Figure 4—video 1*), which could be further used as internal fiducials for correlation with FLM data (*Scher et al., 2021*; *Okolo et al., 2021*; *Klein et al., 2021*).

Using a specimen of HeLa cells stained for mitochondria and lipid droplets, we demonstrate the possibility to combine cryo-FLM with cryo-FIB-SEM volume imaging acquired with SerialFIB. Following correlation in 3DCT, we defined a window for volume imaging spanning two cells (*Figure 4G*). Using the centroids of lipid droplets and microbeads in both imaging modalities to fit an affine transform, we were able to register the two volumes (*Figure 4H*, *Figure 4—figure supplement 2*, *Figure 4—video 3*, 'Data availability'). The root-mean-square residual of the fitted transform was 386 nm, with outliers occurring in clusters, indicating potential local elastic image deformations (*Figure 4—figure supplement 2*). In the present dataset, the alignment of SEM slices was also heavily dominated by surface structures outside of the sliced volume. These features presumably lie on a different incline relative to the imaged slices and thus contribute to the shear in y along z. Furthermore, stretching of the cryo-FLM volume in z was necessary, reminiscent of axial distortion correction factors that are applied in light microscopy to account for compression when the refractive index of the immersion medium (air) is lower than that of the specimen (ice) (*Diel et al., 2020*). In line with a recent study on lipid droplet-based registration between cryo-FLM and cryo-FIB-SEM volumes (*Scher et al.,*

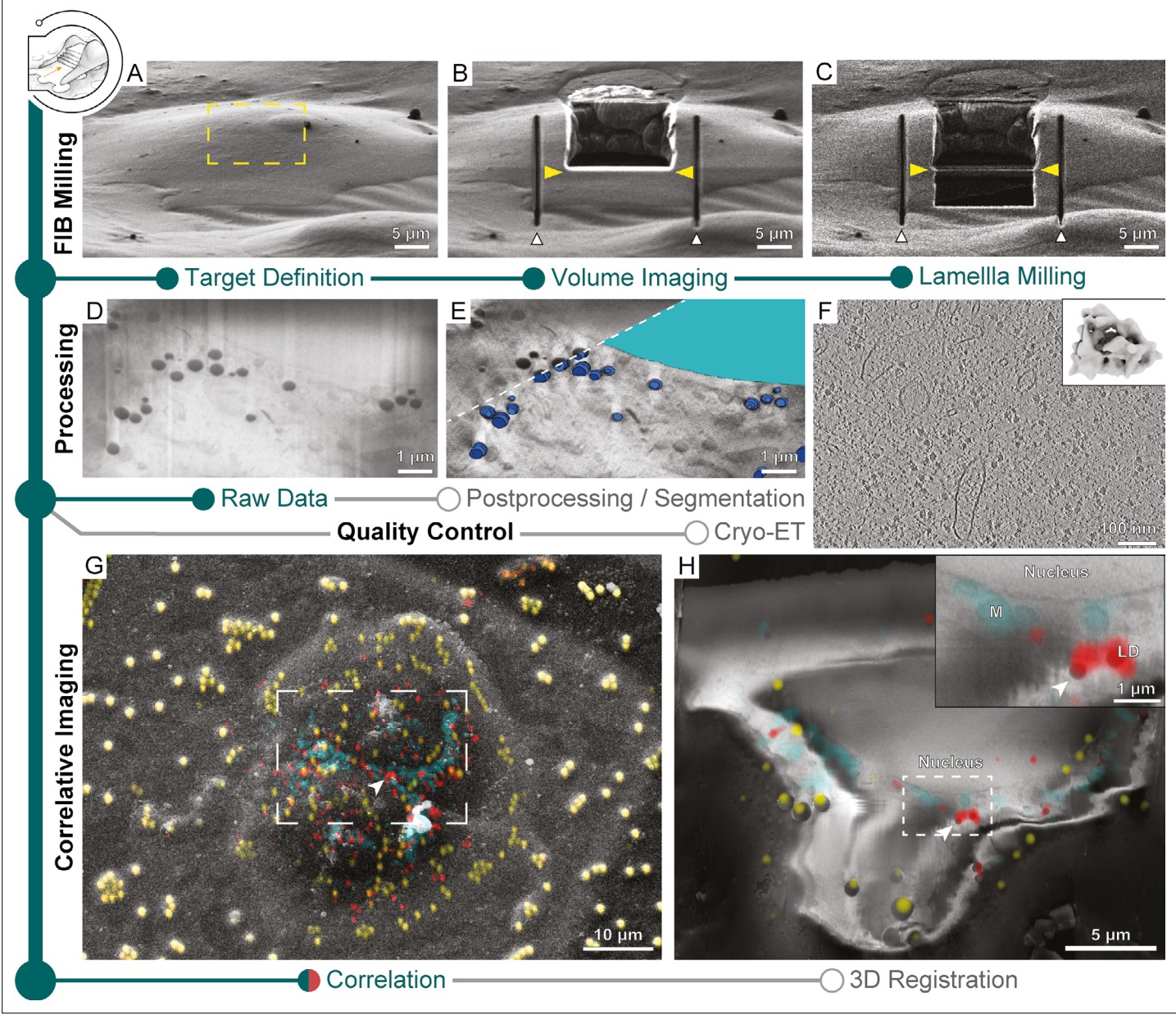

**Figure 4.** Multi-modal 3D cryogenic imaging by focused ion beam-scanning electron microscopy (FIB-SEM) volume imaging and cryo-electron tomography (cryo-ET). Green modules represent tasks performed with SerialFIB; orange, with 3D Correlation Toolbox (3DCT). Gray modules represent tasks performed externally. (**A**) Focused ion beam (FIB) image of a Sum159 breast cancer cell prior to milling. Yellow rectangle indicates the volume for FIB-SEM volume data acquisition. (**B**) FIB image of the cell after FIB-SEM volume imaging. Yellow arrowheads indicate the defined position for lamella preparation. White arrowheads indicate micro-expansion joints prepared after volume imaging. (**C**) FIB image of the final lamella. (**D**) Representative slice of the raw FIB-SEM volume data and (**E**) postprocessed data, overlaid with manual segmentations of lipid droplets (dark blue) and the nucleus (cyan). (**F**) Slice through a tomogram acquired from lamella in (**C**). Inset shows the ribosome subtomogram average from data collected on lamellae prepared by this workflow (two tomograms; 3380 subtomograms; 24 Å resolution). (**G**) SEM view of two HeLa cells overlaid with a fluorescence light microscopy (FLM) maximum intensity projection of lipid droplet (red), mitochondria (cyan), and fiducial bead (yellow) fluorescence signals, determined by correlation in 3D Correlation Toolbox (3DCT). (**H**) Representative slice through the postprocessed FIB-SEM volume overlaid with a 200 nm virtual slice through the affine-transformed 3D-registered fluorescence volumes. White arrowhead points to a pair of lipid droplets, which are displayed at higher magnification in the inset.

The online version of this article includes the following video and figure supplement(s) for figure 4:

**Figure supplement 1.** Serial focused ion beam-scanning electron microscopy (FIB-SEM) volume imaging of *C. reinhardtii* and subsequent image processing adapted from *Spehner et al., 2020*.

**Figure supplement 2.** Point-based affine 3D registration between focused ion beam-scanning electron microscopy (FIB-SEM) and fluorescence

*Figure 4 continued on next page*

*Figure 4 continued*

volumes.

**Figure 4—video 1.** Cryo-focused ion beam-scanning electron microscopy (cryo-FIB-SEM) volume of a Sum159 cell depicting raw and postprocessed data with segmentations of lipid droplets (blue) and the nucleus (cyan).

https://elifesciences.org/articles/70506/figures#fig4video1

**Figure 4—video 2.** Tomographic volume of a Sum159 breast cancer cell related to Figure 4F depicting the cytosol with two vault structures and a vesicle (V).

https://elifesciences.org/articles/70506/figures#fig4video2

**Figure 4—video 3.** Focused ion beam-scanning electron microscopy (FIB-SEM) volume of HeLa cells in Figure 4H, superposed with lipid droplet and bead segmentations and fluorescence volumes transformed.

https://elifesciences.org/articles/70506/figures#fig4video3

*2021*), stretching of the cryo-FLM volume may also have been required due to stage drift in our cryo-FIB system. Overall, our data indicate that non-affine aberrations in cryo-FLM and SEM imaging, not corrected for here, do not deteriorate the registration accuracy beyond the level required for 3D-CLEM-guided lamella preparation. Thus, such extensions of multi-modal imaging could potentially be employed for advanced preparation of targeted lamellae extraction from voluminous samples.

## Lift-out lamella preparation

By rapid cooling at elevated pressures, HPF preserves ultrastructure within 200-µm-thick specimens and thus enables the study of multicellular organisms by cryo-ET. However, as the resultant specimen is embedded in 50–200-µm-thick ice, cryo-FIB lift-out approaches are needed to physically extract material from the bulk specimen for preparation of electron-transparent lamellae.

A time-consuming step in lift-out sample preparations is the generation of trenches that are wide enough to allow access by a gripper-type cryo-micromanipulator and are deep enough such that the lamella captures a meaningful portion of the biological sample. Here, we used SerialFIB for the auto-mated milling of trenches in a high-pressure frozen *Drosophila melanogaster* egg chamber specimen (*Figure 5*). 3DCT can interface with the correlative module in SerialFIB to identify target positions on the HPF planchette by correlation of 2D slices of the cryo-FLM confocal data (*Figure 5A*) with FIB or SEM images (*Figure 5A–C*, *Figure 5—figure supplement 1*). Using the automated procedure, we milled trenches by defining a custom pattern sequence file using the Pattern Designer (*Figure 1—figure supplement 1C*). Due to excessive charging of the HPF sample surface during trench milling, reference image acquisition and all milling steps were performed with 'Drift Suppression,' a function-ality available through the microscope user interface, whereby the electron beam is used to compen-sate for positive charges brought in by the ion beam during milling. Trenches were defined in a horseshoe-shaped arrangement to yield a 20 µm × 20 µm × ~25 µm extractable chunk from the bulk specimen (*Figure 5D*). Milling was performed with the specimen surface perpendicular to the ion beam.

The preparation time of a single position is dependent on the sample and chunk size, and is in the range of 30–60 min. This adds up to ~3–5 hr prior to lift-out for five sites. After this step, an undercut is produced by FIB milling, leaving the chunk attached to the bulk material on one side only. The prepared site can subsequently be approached by a cryo-gripper mounted on a micromanipulator (*Schaffer et al., 2019*), secured between the gripper arms, detached from the bulk by FIB milling and transferred to a specialized carrier grid, termed half-moon grid (*Figure 5E*). Unlike previous studies where the half-moon grid was fashioned into slots prior to lamella deposition (*Schaffer et al., 2019*), here we attached the chunks directly to the grid pins. Lamellae attachment was performed by re-de-position of the grid material onto the lamella while held in place by the gripper, similar to proce-dures described for tungsten needles (*Kuba et al., 2021*). Subsequently, the gripper was moved to a safe position and organometallic platinum was deposited to secure the lamella to the grid pin (*Figure 5E*). Finally, the lamellae were fine-milled manually at a pre-tilt of 10° with respect to the grid plane (*Figure 5F and G* inset), and the grid transferred to the TEM for cryo-ET data acquisition (*Figure 5G*). Tomograms from lamellae prepared with the semi-automated procedures for lift-out site preparation were visually indistinguishable from those prepared via on-grid milling and yield ribosome subtomogram averages of similar quality (*Figure 5H*).

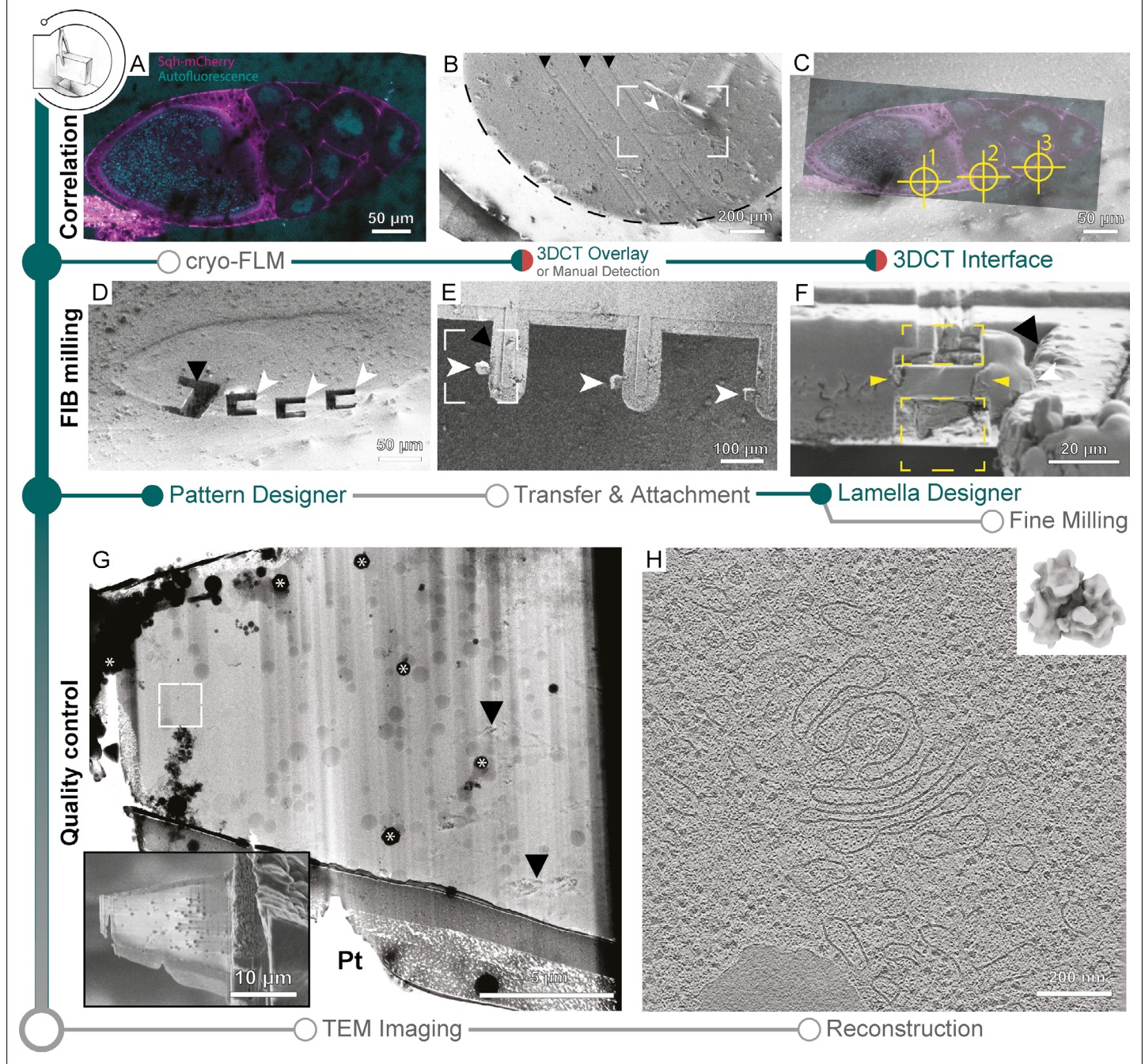

**Figure 5.** Cryo-focused ion beam (cryo-FIB) lift-out from high-pressure frozen (HPF) *D. melanogaster* egg chambers. (**A**) Cryo-fluorescence light microscopy (cryo-FLM) image of an egg chamber expressing Sqh-mCherry (magenta). Autofluorescence (cyan) indicates oocyte and nurse cell nuclei. The anterior of the egg chamber is to the right. (**B**) Scanning electron microscopy (SEM) overview of the sample in (**A**). Black arrowheads indicate knife marks introduced during planing in the cryo-ultramicrotome. White arrowhead indicates the egg chamber. Dashed line outlines the frozen material within the HPF planchette. Frame indicates the area depicted in (**C**). (**C**) Overlay of the cryo-FLM and SEM image of the egg chamber. Crosshairs indicate positions chosen for trench preparation, marked 1–3. (**D**) SEM image of sites prepared for lift-out. Black arrowhead indicates a site produced for FIB-SEM volume imaging to confirm correlation with the fluorescence data, prior to lift-out site preparation (compare *Figure 5—figure supplement 1*). (**E**) SEM overview of the half-moon grid after transfer and attachment of the lift-out specimens. White arrowheads indicate the biological material attached to the grid pins (black arrowhead). Frame indicates the region shown in (**F**). (**F**) FIB image of the lift-out specimen (yellow arrowheads) after attachment to a half-moon grid. Yellow rectangles indicate the milling area for lamella thinning. White arrowhead indicates the lift-out specimen from (**F**). Black arrowhead indicates the edge of the half-moon pin. (**G**) Transmission electron microscopy (TEM) overview of the lamella from (**F**). Inset shows the SEM image of the final lamella prior to transfer to the TEM. White frame denotes the area of the tomogram shown in (**H**). Asterisks indicate ice contamination from transfer, arrowheads indicate reflection artifacts from poorly vitrified local regions. Pt: protective organometallic platinum layer

*Figure 5 continued on next page*

*Figure 5 continued*

at the front of the lamella. (**H**) Single slice through the tomogram collected on the lamella shown in (**G**). Inset shows the *D. melanogaster* ribosome subtomogram average produced from the dataset collected on two lift-out lamellae (8 tomograms; 20,284 subtomograms, 20.8 Å).

The online version of this article includes the following video and figure supplement(s) for figure 5:

**Figure supplement 1.** Correlation for cryo-focused ion beam (cryo-FIB) lift-out from high-pressure frozen (HPF) *D. melanogaster* egg chamber.

**Figure 5—video 1.** Tomographic volume from a *D. melanogaster* lift-out lamellae related to Figure 5H depicting the cytosol with a lipid droplet (LD) and a Golgi apparatus.

https://elifesciences.org/articles/70506/figures#fig5video1

The presented workflow provides an example of a custom task using the pattern sequence file routines for lift-out site preparation based on current state-of-the-art HPF sample carrier, microscope, and grid designs. As the technology evolves, new protocols can be designed and implemented through SerialFIB, enabling the testing of new geometries and milling strategies to further streamline lift-out procedures.

## Discussion

Here, we present an open-source software tool for the automation of cryo-FIB applications, aimed at improving throughput for in situ structural biology. We provide ready-to-use, customizable protocols for a number of use cases. Adjustment of protocols is possible through a GUI, and custom scripts can be executed via a scripting interface. We anticipate that these tools will be valuable to the growing structural cell biology community in supporting the development of additional protocols and advanced multi-modal imaging approaches. While communication with the microscope is based on a proprietary API provided by the microscope manufacturer, functions required for automation, including alignment tasks or transformation of FLM data, are decoupled from the commercial scripting interface. Therefore, the software architecture allows for expansion to other microscope systems by adapting the driver to APIs provided by the respective manufacturer (Appendix 1).

The modularity of the tools presented here makes it possible to adapt the automation to the specific needs of the biological question at hand. However, optimized cryo-grid preparation remains a stringent requirement for successful automated lamellae micromachining workflows. The density and distribution of cells on TEM grids is a crucial factor that determines vitrification quality and milling success. To this end, the concentration of cell in suspensions and the applied volume need to be optimized for each sample type. For adherent cells, the use of micropatterning to direct their seeding helps to improve their positioning on grids for lamella preparation (*Toro-Nahuelpan et al., 2020*). Blotting time and direction also require optimization per sample type. Additional factors should be considered when optimizing for a 3D-CLEM-guided FIB workflow:

1. Mechanical deformation can originate from handling of the specimen, tension within the support due to freezing, or charges introduced by FIB milling or SEM imaging. Two recent hardware solutions have been developed to reduce manual handling of cryo-grids: integration of a fluorescence microscope into the FIB-SEM chamber (*Gorelick et al., 2019*) or the use of shuttles that can be loaded both in the cryo-FLM and cryo-FIB-SEM microscope (*Kuba et al., 2021*). Tension buildup in the grid support during freezing likely arises due to differences in thermal expansion coefficients between the metal mesh and the support film (*Russo and Passmore, 2016*), or from tension in the ice film itself (*Naydenova et al., 2020*). In our experiments, titanium grids with almost continuous $SiO_2$ support appeared to perform better in 3D-targeted lamellae preparations than gold $SiO_2$ grids. Nevertheless, local deformations were evident during FIB imaging and milling regardless of grid type in areas of thin ice. Thus, grids with stable supports should be considered for cellular preparations and ice thickness should be optimized in the vicinity of the milling area. Furthermore, we show that native cellular structures such as lipid droplets that can be stained with live dyes provide a means to identify and validate the 3D positioning of cryo-FLM data corresponding to the produced lamella. Using such structures as fiducials to navigate the lamella during TEM imaging may further assist in the localization of smaller structures for cryo-ET data acquisition.

2. It has been shown for rigid-body registration that the registration error, which describes the offset of a correlated target point from its true position, increases with increasing distance from the fiducials distribution principal axes (*West and Fitzpatrick, 1999*). Thus, an isotropic

distribution of fiducials around the target point is important to minimize the registration error. The fluorescence signal intensity of fiducials and features of interest are also of importance as they determine how confidently the objects can be localized, which also contributes to the registration error. In this respect, 1-μm-sized fluorescent beads are a good choice of fiducials for 3D-CLEM-guided workflows as they are sufficiently large to detect by FIB and SEM imaging and sufficiently bright in cryo-FLM when embedded in ice.

3. Current solutions for diffraction-limited cryo-FLM employ non-immersion (dry) objectives (*Wu et al., 2020*; *Schorb et al., 2017*; *Plitzko et al., 2009*). Thus, refractive index mismatches between the immersion medium (air) and the vitreous specimen are unavoidable. This results in a point spread function that increases in asymmetry with imaging depth (*Li et al., 2019*), as well as axial compressions of the imaged volume with respect to real object (*Diel et al., 2020*). For single-cell specimens, where the cellular volume does not exceed 10 μm in thickness, our cryo-FIB-SEM-based analysis of microbeads and lipid droplets in HeLa cells suggests that affine transforms are sufficient for overcoming potential imaging aberrations and achieving reasonable correlation accuracies for 3D-CLEM-guided milling. We expect that for thicker HPF specimens depth-dependent aberrations may become a limiting factor. While we have not compensated for such aberrations in this work, corrections similar to those performed in single-molecule localization microscopy (*Li et al., 2019*) may provide a solution if the z-position of the air-specimen interface can be identified.

4. Additional errors are introduced when correlating between 3D cryo-FLM data and FIB-SEM volumes. Nonuniform FIB slicing thickness affects the precision with which fiducials can be localized. In the HeLa cell example, FIB milling steps of 100nm were performed, and fiducial centers were defined based on the unweighted center of mass of manually-segmented features. Depending on the homogeneity of the feature throughout slices and imaging contrast, the method of detection and localization should be carefully considered.

Once samples are optimized, on-grid lamella milling can become a routine task. Depending on the sample thickness, a single grid of large cells (e.g., HeLa), two or more grids for smaller cells (e.g., *Escherichia coli, Saccharomyces cerevisiae*) can be processed in a single day by automated lamella preparation. However, fewer lamellae can be generated through cryo-FLM targeting or following high-quality cryo-FIB-SEM volume imaging owing to the additional time consuming correlation or imaging steps.

Understanding the caveats of 3D-CLEM at cryogenic conditions through the use case of single-cell samples paves the way for more challenging endeavors, namely identifying and targeting subcellular structures in HPF multicellular specimens. The increasing cooling runtimes of cryo-FIB-SEM microscopes paired with customized automation will enable shifting time-consuming steps of cryo-FIB-SEM volume imaging to overnight operation. While conventional on-grid milling already benefits from the higher throughput made possible by overnight runs and lower in-chamber contamination (*Tacke et al., 2021*), the impact on successfully targeted cryo-FIB lift-out is expected to be also significant. Currently, cryo-FLM-guided site-specific preparation for in situ lift-out relies on manual identification of global features from cryo-FLM data at the surface of the sample for rough registration. Microbead fiducials, if introduced before cryo-fixation, would be buried within the bulk specimen and therefore preclude precise 3D registration. Cryo-FIB-SEM volume imaging prior to cryo-FIB lift-out should allow identification of fiducials in the bulk specimen, whether they be fluorescent microbeads or biological landmarks, and subsequent precise registration with the 3D cryo-FLM data. This opens the possibility of using fluorescence signal in combination with cryo-FIB-SEM volume imaging to refine targeting for lift-out approaches once on-the-fly postprocessing and automated segmentation of the cryo-FIB-SEM volume data become available (*Tian et al., 2015*; *Berg et al., 2019*; *Darrow et al., 2017*). In this case, automated overnight operation can increase both throughput and success rate, while the experimenter can focus solely on error-prone steps such as the lift-out procedure or fine milling of the extracted lamellae. The in situ lift-out approach is an established technique in the material sciences for room temperature applications. With increasing operation experience at cryogenic conditions, further automation will increase throughput of the remaining steps in this workflow, including undercut milling, lamella extraction, transfer, and attachment.

The biological research question at hand will dictate the use case. Projects will benefit from automation due to the higher throughput for in situ structural biology, improved 3D targeting for on-grid lamella generation, and cutting-edge preparation from voluminous multicellular specimens obtained along with valuable contextual information from FIB-SEM volume imaging. The improved practical

application of multi-modal imaging under cryogenic conditions through automation holds the potential to accelerate the acquisition of biological insights.

# Materials and methods

## Key resources table

| Reagent type (species) or resource | Designation | Source or reference | Identifiers | Additional information |
|---|---|---|---|---|
| Cell line (*Homo sapiens*) | HeLa Kyoto | Hyman Lab, MPI-CBG | RRID:CVCL_1922 | |
| Cell line (*H. sapiens*) | Sum159 | Walter and Farese Lab, Harvard T.H. Chan School of Public Health | RRID:CVCL_5423 | |
| Strain, strain background (*Saccharomyces cerevisiae*) | Ede1-eGFP | Wilfling lab, MPI for Biophysics | FWY0153 | MATα, his3-Δ200, leu2-3,2-112, lys2-801, trp1-1(am), ura3-52, yap1801Δ::kanMX4, yap1802Δ::hphNT1, apl3Δ::HIS3MX6, Ede1::EGFP::TRP1, atg15Δ::natNT2, atg19Δ::URA3 |
| Strain, strain background (*Emiliania huxleyi*) | Eh1 | Vardi Lab, Weizmann Institute of Science | | Isolated in June 2018 at Espegrend, Norway |
| Strain, strain background (*Chlamydomonas reinhardtii*) | mat3-4 | Chlamydomonas Resource Center, University of Minnesota, MN | CC-3994 | |
| Strain, strain background (*C. reinhardtii*) | CW15 | Chlamydomonas Resource Center, University of Minnesota, MN | CC-400 | Reference cell-wall deficient WT strain of Chlamydomonas |
| Genetic reagent (*Drosophila melanogaster*) | sqh-mCherry | Bloomington Drosophila Stock Center | FLYB: FBst0059024 | FlyBase genotype: w*; P{sqh-mCherry.M}3 |
| Other | BODIPY 558/568C12 | Thermo Fisher Scientific | D3835 | Lipid droplet live stain (1:2000) |
| Other | Mito Tracker Green (490/516) | Thermo Fisher Scientific | M7514 | Mitochondria live stain (1:2000) |
| Other | Carboxylate-Modified Microspheres, 1.0 µm, crimson fluorescent (625/645) | Thermo Fisher scientific | F8816 | Fluorescence bead fiducials for 3D correlation (1:30) |
| Software, algorithm | SerialFIB | https://github.com/sklumpe/SerialFIB (copy archived at swh:1:rev:0eaaaf66afa2d803440cea18af85c444df10478f, *Klumpe, 2021*) | | This study |
| Software, algorithm | AutoScript4 | Thermo Fisher Scientific | | https://www.thermofisher.com/de/de/home/electron-microscopy/products/software-em-3d-vis/autoscript-4-software.html |

*Continued on next page*

*Continued*

| Reagent type (species) or resource | Designation | Source or reference | Identifiers | Additional information |
|---|---|---|---|---|
| Software, algorithm | LAS X | Leica Microsystems | | https://www.leica-microsystems.com/products/ microscope-software/p/leica-las-x-ls/ |
| Software, algorithm | Fiji | https://imagej.net/software/fiji/ | | |
| Software, algorithm | 3DCT | https://3dct.semper.space/index.html | | |
| Software, algorithm | SerialEM | https://bio3d.colorado.edu/SerialEM/ | | |
| Software, algorithm | Warp | http://www.warpem.com/warp/ | | |
| Software, algorithm | TOMOMAN | https://github.com/williamnwan/TOMOMAN | | |
| Software, algorithm | MotionCorr2 | https://emcore.ucsf.edu/ucsf-software | | |
| Software, algorithm | NovaCTF | *Turoňová et al., 2017b* | | |
| Software, algorithm | CTFFIND4 | https://grigorefflab.umassmed.edu/ctffind4 | | |
| Software, algorithm | IMOD | https://bio3d.colorado.edu/imod | | |
| Software, algorithm | pyTOM | https://pytom.sites.uu.nl/ | | |
| Software, algorithm | RELION | https://www3.mrc-lmb.cam.ac.uk/relion/index.php/Main_Page | | |
| Software, algorithm | STOPGAP | *Wan, 2020* | | |
| Software, algorithm | UCSF ChimeraX | https://www.rbvi.ucsf.edu/chimerax/ | | |
| Other | Cryo-Gripper | Kleindiek Nanotechnik | | https://www.nanotechnik.com/cryoliftout.html |

## Mammalian cell culture and cryo-grid preparation

HeLa Kyoto cells were cultured in DMEM (Thermo Fisher Scientific), supplemented with 10% (v/v) fetal bovine Serum (FBS; Biochrom) and 100 mg/mL penicillin-streptomycin (Thermo Fisher Scientific). Sum159 cells were cultured in DMEM/F12, glutaMAX media (Life Technology) supplemented with 5% FBS, 1 µg/mL hydrocortisone (Sigma), 5 µg/mL insulin (Cell Applications), 10 mM HEPES (pH 7.4), and 100 mg/mL penicillin-streptomycin (Thermo Fisher Scientific). Cells were maintained in TC-25 flasks (Thermo Fisher Scientific) at 37°C under 5% $CO_2$. Both lines tested negative for *Mycoplasma* by PCR (Eurofins Genomics). For EM grid seeding, cells at 60–70% confluence were treated with 0.05% trypsin-EDTA (Fisher Scientific) for 2 min at 37°C, resuspended in 1 mL media, passed through a 35 µm mesh Cell Strainer Snap Cap (Corning), and counted. 200,000 cells were seeded into a 35 mm ibidi dish (ibidi) containing up to eight micropatterned EM grids and 2 mL media, to reach an average surface density of $2 \times 10^4$ cells/$cm^2$. Gold or titanium grids, 200 mesh, with 12-nm-thick holey R1/4 and R1.2/20 $SiO_2$ film (Quantifoil) were used for the experiments. Grids were passivated, micropatterned, and fibronectin-treated according to *Toro-Nahuelpan et al., 2020*. Grids were incubated for 2 hr with HeLa cells or 1 hr with Sum159 cells at 37°C under 5% $CO_2$, and then transferred to a new

dish with media for further incubation. Grids were plunge-frozen into liquid ethane at –185°C 4–18 hr post-transfer, depending on the desired cell density per square. Plunge-freezing was performed with a Leica GP EM at 37°C and 75% humidity, and 1 s blot time for all grid types. For correlative experiments, grids were incubated with MitoTracker Green FM (Thermo Fisher Scientific) and BODIPY 558/568 (Thermo Fisher Scientific) at 1:2000 dilutions for 20 min prior to plunge freezing. With the grid mounted on the plunger, 4 μL Crimson Microspheres, 1.0 μm in diameter (Thermo Fisher Scientific), diluted 1:30 in PBS, were applied to the grid from the side containing cells.

## Coccolithophore cryo-grid preparation

*E. huxleyi* strain Eh1 was isolated at the Espegrend Marine Research Field Station, Norway, and provided by Prof. Assaf Vardi from the Weizmann Institute of Science. Cells were grown in artificial seawater, supplemented with f/2 nutrient recipe to late exponential phase, under 16 hr/8 hr light/dark cycles at 18°C. In order to induce calcification, the external mineral coccolith shell was removed by treating with 20 mM EDTA at pH 8. The de-calcified cells were moved to a calcium-depleted medium (100 μM $Ca^{2+}$) for 12 hr. Then $Ca^{2+}$ was supplemented to the normal seawater level of 10 mM. This lag time in a calcium-depleted medium allowed the cells to resume mineralization in a synchronized fashion upon $Ca^{2+}$ repletion. Cells were centrifuged 3 hr after re-calcification at 2000 × *g* for 3 min to increase their concentration for plunge freezing. A volume of 4 μL cell suspension at a concentration of 3.07–3.5 × $10^7$ cells/ml was directly applied to glow-discharged holey carbon R2/1 Cu 200 mesh grids (Quantifoil). After 5 s back-side blotting, plunge-freezing was carried out with a Leica GP EM at 18°C and 90% humidity.

## Budding yeast cryo-grid preparation

The yeast strain used was *S. cerevisiae* ΔYAP1801 ΔYAP1802 ΔAPL3 EDE1-eGFP ΔATG15 ΔATG19. Yeast cultures were inoculated from overnight cultures started from colonies and grown in YPD media at 30°C to an $OD_{600}$ of 0.8. For grid preparation, 4 μL of cell suspension was applied to holey carbon R2/1 Cu grids (Quantifoil) and plunge-frozen on a Vitrobot Mark IV (Thermo Fisher Scientific) at blot force of 10, blotting time of 10 s, temperature 30°C, and humidity of 90%.

## *Chlamydomonas* cryo-grid preparation

*C. reinhardtii mat3-4* and *CW15* strains were grown with 100 rpm agitation in Tris-Acetate-phosphate Medium (TAP Medium) at room temperature and normal atmosphere under continuous illumination (40 μE white light). Log phase cultures were centrifuged at 2000 × *g* for 2 min to concentrate the cells 10 times and 4 μL was directly applied to a glow-discharged holey carbon R2/1 Cu 200 mesh grid (Quantifoil). Plunge-freezing was performed using a VitroBot Mark 4 (Thermo Fisher Scientific) with Teflon sheets on both sides and blotting from the back. Blotting time was 10 s with a blotting force of 10 at 90% humidity.

## Cryo-FIB lamella preparation

Lamellae were prepared as described in *Schaffer et al., 2017* on a dedicated Aquilos dual-beam (FIB-SEM) microscope equipped with a cryo-transfer system and a cryo-stage (Thermo Fisher Scientific). The AutoScript 4 software is commercially available from Thermo Fisher Scientific.

Cryo-grids with plunge-frozen cells were clipped into autogrids modified with a cut-out for FIB milling, mounted on a shuttle with a 45° pre-tilt (Thermo Fisher Scientific), and transferred into the dual-beam microscope. During FIB operation, samples were kept in high vacuum (<4 × $10^{-7}$ mbar) and at constant liquid nitrogen temperature using an open nitrogen circuit, 360° rotatable cryo-stage. Samples were first coated with organometallic platinum using the gas injection system (GIS, Thermo Fisher Scientific) operated at 28°C, 10.6 mm stage working distance and 8–11 s gas injection time to protect the targeted lamella front during FIB milling. Subsequent sputter coating with platinum (10 mA, 20 s) was performed to improve sample conductivity. Appropriate positions for lamella preparations were identified, registered, and the eucentric/coincidence height determined and refined manually or in the MAPS 3.3 software (Thermo Fisher Scientific). Lamellae were prepared at a stage tilt angle of 16–20° using a gallium ion beam at 30 kV. Lamella preparation protocols, including FIB currents and milling times for each cell type, are summarized in *Supplementary file 1*. After fine

milling, grids were sputter-coated with platinum (10 mA, 5 s), transferred into grid boxes, and stored in liquid nitrogen.

## Correlative lamella preparation and analysis

Clipped grids were imaged on a prototype Leica cryo-confocal microscope based on the Leica TCS SP8 system, equipped with a cryo-stage (*Schorb et al., 2017*), ×50 objective, NA 0.90, and two HyD detectors. On this system, the full width at half maximum of the signal profiles for 200 nm Tetraspeck Microspheres (Thermo Fisher Scientific) excited at 488 nm and detected at 500–550 nm with a pinhole size of 1 Airy unit and voxel size of 87 nm × 87 nm × 490 nm measures 660 nm in x,y and 1130 nm in z. Widefield overview montages were acquired in brightfield and fluorescence, and grid squares of interest identified. Z-stacks of HeLa cells stained for mitochondria and lipid droplets and plunge-frozen with fluorescent beads were obtained by 488 nm laser excitation, detecting at 500–545 nm (BODIPY), followed by 552 nm excitation, detecting at 561–630 nm (MitoTracker) and 673–731 nm (microbeads), with a xy-pixel size of 109–178 nm and z-step size of 331–342 nm. Z-stacks were aligned in XY using the MultiStackReg ImageJ (*Busse, 2021*; *Thévenaz et al., 1998*) plugin to counter drifts in XY during z-stack acquisition, and then deconvolved using the Lightning module of the Leica LAS X software. In the FIB-SEM microscope, grid squares were identified by overlay of SEM and light microscopy overview images. Coincidence heights were found for each site of interest. SEM and FIB images, encompassing the whole grid square, were acquired for each site. Correlations were established using 3DCT, and the FIB image and 3DCT output were imported into SerialFIB to define lamella positions. Automated rough and fine milling were performed as above. Lamellae were sputter coated as described. Masked fluorescence projections were generated using 3DCT, overlaid onto the SEM image in Fiji (*Schindelin et al., 2012*), and during a TEM session registered in SerialEM with the lamella TEM montage to guide the setting up of positions for tilt series acquisition. For analysis, registrations between SEM images and TEM montages were refined using BigWarp (*Bogovic et al., 2016*) in Fiji. Fluorescence 'virtual slices' corresponding to different heights in the final FIB image were created using the masking utility in 3DCT, projected onto the SEM and the correlated TEM image of lamellae, to identify the fluorescence plane that corresponds best to the structural features in the lamellae (lipid droplets and microbeads). Local mechanical deformation was assessed by elastic registration of FIB images before and after milling using bUnwarpJ (*Arganda-Carreras et al., 2006*). Broken squares, contaminants, and the site of milling were masked out for this analysis.

## New 3DCT features

1. For bead detection in the SEM image, pixels are scored based on three measures. First, the image is cross-correlated with a Gaussian kernel, where sigma equals a user-specified bead radius. Second, bead edges are detected with a Canny filter and Hough transform. Pixels more likely to be centers of circles of the given radius are scored higher. The radius is based on the user-provided value and an adjustable multiplier. Third, a soft mask is generated by thresholding of the image with a method of choice, followed by dilation and Gaussian blurring. Scores from all measures are multiplied to obtain a final score. Top peaks are selected and returned as a binary image where peak positions are marked for overlay onto the SEM image in the GUI.

2. Based on the generated FLM-SEM correlation, fiducial positions in the FIB image are semi-automatically predicted. This is achieved by first calculating a plane that is closest to all fiducials used in the SEM correlation. The user then selects a point in the SEM image on this plane. Rotation around this point by the angle between the electron and ion beam places the fiducials on the FIB image. Finally, fiducial positions are translated according to a matching point in the FIB image selected by the user to correct for beam shifts and deviation from coincidence height. Fiducial positions are returned in a comma-separated values file that can be imported into the GUI.

3. To create a 3D mask in the fluorescence volume that matches the position of the generated lamella, a plane is defined by the user by selecting a point in the FIB image and the calculated transform between FLM and FIB images. Given a user-defined polygon in the SEM image, which marks the boundary of the lamella, lines passing through the corner points of this polygon and normal to the SEM image are drawn. The intersection between these lines and plane defined gives the 2D contour of the lamella in 3D space. An additional point placed by the user on the lamella in the SEM image is used to identify the interior of the lamella. A slice of the specified

thickness is drawn centered on the contour calculated. The resulting mask can then be applied to the FLM volume during projection generation.

4. To generate a projection based on a calculated affine transform, FLM volumes are transformed by inverse mapping with linear interpolation, then the maximum intensity projection is taken. This computation is multi-threaded as projections are calculated in patches. To minimize computation, the minimal Z-range that needs to be evaluated to cover the transformed volume is determined before inverse mapping of the FLM data for each patch. Patch size and number of CPUs employed are tuned according to the capacity of the support computer. With this, unbinned fluorescence volumes can be projected onto high-resolution SEM or FIB images during a FIB-SEM session to help guide the setting up of lamella positions in SerialFIB. All features described can be accessed through a PyQt5 graphical interface from command line or from within 3DCT, which has also now been ported to Python 3.

## Cryo-FIB-SEM volume imaging and analysis

For cryo-FIB-SEM volume imaging, samples were prepared as described above. For Sum159 and HeLa cells, SEM images (3072 × 2048 px, pixel size of 10.377 nm and 19.271 nm, respectively) were recorded at 5 kV and 50 pA with a dwell time of 1 µs using the Everhart–Thornley detector (ETD), with line integration of 16 for noise reduction, resulting in a dose of 0.464 e/Å$^2$ and 0.134 e/Å$^2$ per image, respectively (*Goggin et al., 2020*). Milling was performed at 30 kV and 0.5 nA for 10–20 s per step using 'rectangular cross-section' patterns. Ion beam section thickness was 100 nm, which resulted in step-wise ablation of total volumes of 14.3 µm × 2.0 µm × > 13 µm for Sum159, and 25.9 µm × 6.2 µm × > 16 µm for HeLa. *C. reinhardtii* cells were imaged at 3 kV and 13 pA with a dwell time of 0.25 µs, line integration of 64, at a pixel size of 3.4 nm, and 6144 × 4,096 resolution, corresponding to a dose of 1.12 e/Å$^2$ per image, using the ETD, as well as T1 and T2 in-lens detectors. A stage potential was applied in the OptiPlan mode. Milling was performed at 30 kV and 0.3 nA for 60 s with slices of 50 nm.

For volume image analysis, the SEM stack was cropped to the size of the milled area using Fiji (*Schindelin et al., 2012*). Curtaining from ion beam milling was reduced by wavelet decomposition and Gaussian blurring of the vertical component (sigma = 6) using pywt (*Spehner et al., 2020*). Charging was compensated by Gaussian blurring (sigma = 35) and subsequent image erosion in two steps to create a mask for image subtraction. The script to perform these tasks is available on the GitHub repository (see 'Data availability'). Image contrast was further enhanced by limited adaptive histogram equalization in Fiji (CLAHE, *Zuiderveld, 2014*) with a slope of 3 pixels resulting in restoration of details. Images were aligned using the SIFT algorithm in Fiji (*Schindelin et al., 2012*) and stretched in Y by a factor of 1/sin 52° to compensate for foreshortening due to the angle between ion and electron beams. Segmentations were prepared manually in Amira 2020.2 (Thermo Fischer Scientific).

For point-based registration between FIB-SEM and cryo-FLM volumes, centroids of segmented lipid droplets and beads were extracted by connected-component labeling in MATLAB (MathWorks), and by iterative 1D and 2D Gaussian fitting in 3DCT as described (*Arnold et al., 2016*), respectively. Points were matched in 3DCT using a 2D projection of the segmentation. To avoid local minima, the affine transform relating FLM points to FIB-SEM points was fitted multiple times in Python using (1) the RANSAC approach implemented in OpenCV; (2) L-BFGS-B-based local minimization with different starting positions generated by the TEASER algorithm (*Heng et al., 2020*); and (3) basin-hopping coupled with L-BFGS-B-based local minimization. The FLM volumes were transformed according to the affine transform and superposed with the FIB-SEM volume in Amira for visualization. Python scripts for the affine registration and volume transformation are available on GitHub (see 'Data availability'). Cryo-FLM volumes for the HeLa cells were acquired with a voxel size of 110 × 110 × 342 nm prior to volume imaging.

## Fly husbandry

*D. melanogaster w[*]; P{w[+ mC]=sqh-mCherry.M}3* (FlyBase ID: FBst0059024; *Martin et al., 2008*), expressing an mCherry-tagged Sqh protein (myosin regulatory light chain) under the control of the *sqh* endogenous promoter, were obtained from the Bloomington Drosophila Stock Center (BL-59024) and maintained at 22°C on standard cornmeal agar. Flies were transferred into a fresh vial supplemented with yeast paste 24 h prior to dissection of egg chambers for HPF.

## HPF and cryo-FIB lift-out

*D. melanogaster* egg chambers were dissected from ovaries in Schneider's medium and HPF (HPM010, Abrafluid) in the 100 µm cavity of gold-coated copper Type A carriers (Engineering office M. Wohlwend) using Schneider's medium containing 20% Ficoll (70 kDa) as filling medium. HPF carriers were soaked in hexadecene and blotted on filter paper prior to freezing. Cryo-planing of approximately 40 µm of the surface was performed in a cryo-microtome (EM UC6/FC6 cryo-microtome, Leica Microsystems) using a 45° diamond trimming knife with a clearance angle of 6° (DiAtome). The sample was glued to a custom 3D-printed shuttle after the Leica design (*Kuba et al., 2021*) using cryo-glue (2:3 ethanol:iso-propanol mixture) for cryo-confocal imaging with a Leica TCS SP8 equipped with Leica EM cryo-CLEM stage. Imaging was performed using a HC PL APO ×50/0.9 DRY objective in fluorescence mode using a 552 nm excitation laser at 17% total laser strength and 488 nm excitation at 6.8% total laser strength, corresponding to roughly 0.94 mW and 0.38 mW, respectively (based on measured laser intensity values). Detection channels were 650–751 for mCherry and 495–545 nm for autofluorescence using HyD detectors at a pinhole size of 2 Airy units. Z-stacks of 38 µm from the surface were collected using a step size of 1 µm. Fluorescence signal decreased significantly with imaging depth. Targeting of positions for automated trench milling was based on the surface topology of the sample produced by the cryo-planing. Positions were chosen based on registration with cryo-FLM data in 3DCT, which allowed the identification of regions within the egg chamber.

For lift-out experiments, an Aquilos dual-beam FIB-SEM microscope (Thermo Fisher Scientific) was equipped with a Kleindiek MM3A-EM micromanipulator and a cryo-gripper head (Kleindiek Nano-technik) cooled by copper wires attached to the microscope anti-contaminator. Standard 45° pre-tilt shuttles were used for accommodating both HPF carrier and Auto-Grid clipped receiver grid on an Aquilos stage that was modified by removing ~2 mm of material of the stage using a file to allow the cryo-gripper to reach the sample surface. OmniProbe 4-post Cu half-grids were used as receiver grids (Plano EM). After specimen loading, platinum GIS deposition was performed for 10 s at 27°C with the GIS needle distance at a stage working distance of 10.6 mm. Samples were sputter coated with platinum in the chamber (10 mA, 15 s). For all trench milling steps and SerialFIB session setup, drift suppression using the SEM was applied to compensate positive charges brought in by FIB milling. The stage was rotated by 180° relative to loading position and tilted to 7° to orient the sample surface perpendicular to the FIB. Trench milling was performed with 'regular cross-section patterns' at 30 kV, 1 nA creating a C-shape using SerialFIB (*Figure 5*). The milling included two patterns of 40 µm × 15 µm symmetrically arranged around a 20 µm block. Together with a third pattern, 10 µm × 40 µm was generated at an offset of 15 µm to the left and perpendicular to the previous two, yielding a 20 µm × 20 µm block for lift-out. Milling time per position was 30 min. In total, five positions were prepared, resulting in a milling preparation time of 2.5 hr. Undercuts were performed manually by tilting the stage to 43° and employing 'cleaning cross-section' patterns of 1 µm height, 22 µm width, and 15 µm depth at ~7 µm distance from the sample surface. Lift-out was performed with the micro-gripper oriented parallel to the FIB angle and perpendicular to the sample surface. The produced sample chunks were approached by the gripper. After contact of both tweezer arms with the sample, the chunk was milled loose from the bulk material with 'regular cross-section patterns' at 1 nA with a pattern of 20 µm in length, 1 µm in width until extraction from the bulk was possible. Subsequently, the microgripper was raised until it was ~400 µm above the sample surface. The stage was moved down by ~15 mm into safe distance from the gripper and subsequently moved to the receiver grid position while keeping the Z-height locked. Then, the stage was lifted to the appropriate Z-height of the receiver grid. The microgripper was moved to the post of the TEM half-grid until achieving contact of the sample. 'Regular cross-section' patterns at height of 2 µm and width of 6 µm were used to attach the sample to the post by re-deposition of the post material onto the sample. Subsequently, the gripper was opened, releasing the chunk that has been welded to the pin, and moved to a safe position. This is achieved by lifting the gripper until it is 400 µm above the sample surface and then moving the stage down by 15 mm, making enough room to safely move it to the park position. This procedure can be performed for all four pins. Finally, deposition of platinum GIS for 3 × 30 s in 3 min intervals at a stage working distance of 9 mm was used to strengthen the sample attachment to the pin. Lamella production was done manually with sequentially reduced FIB currents of 3 nA to 5 µm lamella thickness, 1 nA to 4 µm, 0.5 nA to 3 µm, 0.3 nA to 2 µm, and 0.1 nA to 1 µm using 'regular cross-sections' patterns. Subsequent fine milling was performed using 'regular patterns' at 0.1 nA to

600 nm and 50 pA to 200 nm. The FIB divergence was compensated by over- and under-tilts of 1° (*Schaffer et al., 2017*). The lamella was milled until contrast started fading in SEM imaging at 3 kV, 13 pA.

### Cryo-electron tomography

Cryo-ET data was acquired on a Titan Krios (Thermo Fisher Scientific) at 300 kV, equipped with a K2 Summit direct detection camera (Gatan) operating in dose fractionation mode and a Quantum post-column energy filter (Gatan). Autogrids containing lamellae were loaded such that the axis of the pre-tilt introduced by FIB milling was aligned perpendicular to the tilt axis of the microscope. At an EFTEM magnification of 42,000 and a resulting pixel size of 3.37 Å or 3.52 Å, up to 10 tilt series were collected on a single lamella in low-dose mode using SerialEM (*Mastronarde, 2005*). Starting from the lamella pre-tilt, images were acquired at 2.5–4.5 μm underfocus, in 2° increments between ±62° using a dose-symmetric tilt scheme (*Hagen et al., 2017*). The total dose was around 130 e⁻/Å² with a constant electron dose per tilt image. Tilt series of *E. huxleyi*, *C. reinhardtii*, and *S. cerevisiae* were collected with a 70 μm objective aperture. Tomograms of Sum159 and HeLa cells were acquired with a Volta phase plate (VPP, Thermo Fisher Scientific) with prior conditioning for 6 min.

### Tomogram reconstruction

Data were preprocessed in Warp using movie frame alignment to compensate for beam-induced movement, CTF estimation, and tilt stack sorting (*Tegunov and Cramer, 2019*) or TOMOMAN (*Wan, 2021*), employing MotionCorr2 (*Zheng et al., 2017*), NovaCTF (*Turoňová et al., 2017a*), and CTFFIND4 (*Rohou and Grigorieff, 2015*). In etomo (IMOD software package; *Mastronarde and Held, 2017*), four times binned tilt images (movie sums) were aligned using patch tracking and tomograms reconstructed via weighted back projection.

### Subtomogram analysis

Coordinates of ribosomes in Sum159 cells were determined in 4× binned tomograms by template matching utilizing the pyTOM toolbox (*Hrabe et al., 2012*), using a down-sampled human 60S large ribosomal subunit (Gaussian filter with sigma 2, EMDB-2938) as a template and a spherical mask with 337 Å diameter. After manual selection, 4378 and 3380 ribosomal particles were localized in tomograms from standard on-grid FIB-milled lamellae (four tomograms) and lamellae prepared after FIB-SEM volume imaging (two tomograms), respectively. Subtomograms and corresponding CTF models were reconstructed in Warp with a box size of 140 px, a pixel size of 3.37 Å, and assuming a 350 Å particle diameter for normalization. Subtomogram alignment and averaging was performed in RELION (version 3.0.7, *Zivanov et al., 2018*). Particles from both datasets were pooled to generate an initial average via 3D classification into one class with the human 80S ribosome (EMDB-2938), low-pass-filtered to 60 Å, as reference. The resulting average was 3D-refined. Then, particles were

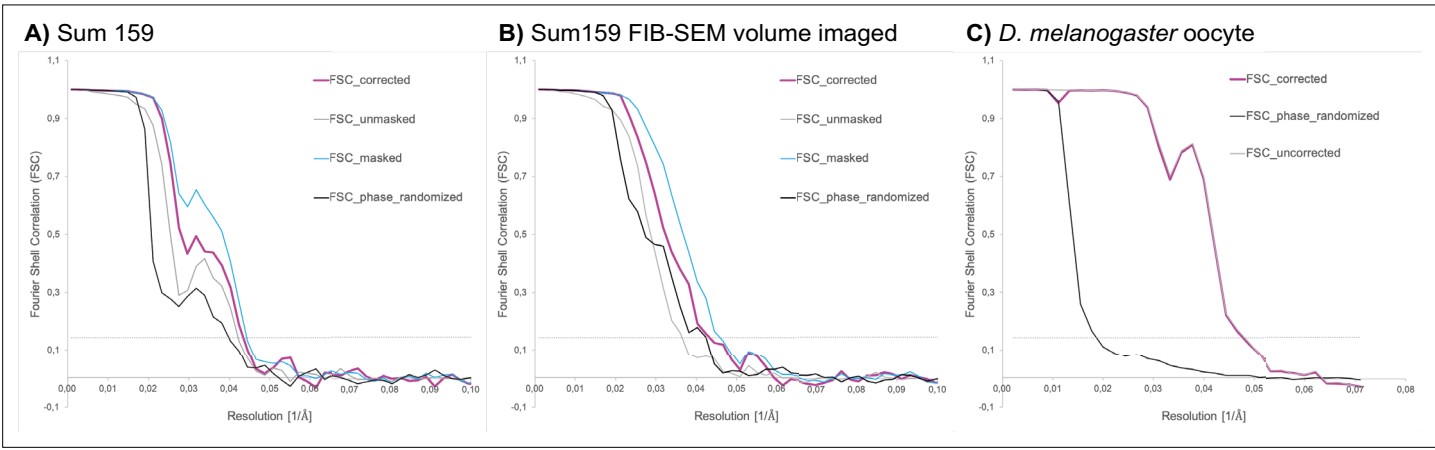

**Figure 6.** Fourier shell correlation (FSC) curves for ribosome subtomogram averages from (**A**) Sum159 cells, (**B**) Sum 159 cells after focused ion beam-scanning electron microscopy (FIB-SEM) volume imaging, and from (**C**) a *D. melanogaster* egg chamber. FSC threshold at 0.143 indicated by gray dotted line.

regrouped based on their origin, that is, from standard FIB-milled lamellae or from lamellae prepared after FIB-SEM volume imaging. Both particle groups were refined separately starting from the alignments of the pooled average and its density as reference. Resolutions of the two final reconstructions were calculated via Fourier shell correlation (FSC) of two half-maps, and the reconstructions were filtered in the RELION postprocessing to their respective resolution (24 Å at 0.143 FSC; *Figure 6*).

*Drosophila* ribosomal particle positions were determined in 4× binned tomograms by template matching in STOPGAP using a down-sampled reference from a previously determined structure of the *D. melanogaster* ribosome (EMD-5591). In total, 20,284 ribosomal particles from eight tomograms were picked. Particles were extracted with a box size of 64 px, at a pixel size of 7.04 Å. Subsequent subtomogram alignment and averaging was performed in STOPGAP (*Wan et al., 2020*). The resolution was calculated via Fourier shell cross-correlation of half-maps to be 24.0 Å (0.5 FSC) and 20.8 Å (0.143 FSC) (*Figure 6*).

Subtomogram averages were visualized with the UCSF ChimeraX package (*Pettersen et al., 2021*).

## Python packages

Implementations were done in Python3. Packages used in this work were numpy (*Harris et al., 2020*), scipy (*Virtanen et al., 2020*), scikit-image (*van der Walt et al., 2014*), open-cv (*Bradski, 2000*), tiffile (*Gohlke, 2021*), psutil (*Rodola, 2020*), (*PyQt5, 2021*), pickle (*Van Rossum, 2020*), and matplotlib (*Price-Whelan et al., 2018*).

## Acknowledgements

We are grateful to the Walther and Farese lab (Harvard TH Chan School of Public Health) for kindly providing Sum159 breast cancer cells, to the Hyman lab (MPI-CBG) for the HeLa cells, Wioleta Dudka for preparation of HeLa cell grids, Zohar Eyal and Assaf Gal (Weizmann Institute of Science) for the *E. huxleyi* grids, and Wojciech Wietrzynski and Ben Engel (Helmholtz Pioneering Campus) for the *C. reinhardtii* grids. We thank members of the Mahamid group, Martin Schorb (Electron Microscopy Core Facility, EMBL), the EMBL cryo-EM platform, Thomas Hoffmann, Florian Beck, Anna Bieber, Cristina Capitanio, and Sagar Khavnekar for invaluable input and support. The cryo-confocal laser scanning microscope (TCS SP8-Cryo CLEM) was developed in collaboration with Leica Microsystems. HKHF was supported by a fellowship from the EMBL Interdisciplinary Postdoctoral Program (EI3POD) under Marie Skłodowska-Curie Actions COFUND (664726). SK was supported by the International Max Planck Research School for Molecular and Cellular Life Sciences. JM acknowledges funding from the EMBL, the European Research Council (ERC 3DCellPhase⁻ 760067), and iNEXT-Discovery project (871037).

## Additional information

### Competing interests

Jürgen M Plitzko: holds a position on the advisory board of Thermo Fisher Scientific. The other authors declare that no competing interests exist.

### Funding

| Funder | Grant reference number | Author |
| --- | --- | --- |
| European Research Council | 760067 | Julia Mahamid |
| European Research Council | 871037 | Julia Mahamid |
| European Molecular Biology Laboratory | Interdisciplinary Postdoctoral Program COFUND 664726 | Herman KH Fung |
| Max Planck Institute Magdeburg | | Sven Klumpe |

| Funder | Grant reference number | Author |
|---|---|---|
| Marie Sklodowska-Curie Actions | Interdisciplinary Postdoctoral Program COFUND 664726 | Herman KH Fung |

The funders had no role in study design, data collection and interpretation, or the decision to submit the work for publication.

## Author contributions

Sven Klumpe, Herman KH Fung, Sara K Goetz, Conceptualization, Data curation, Formal analysis, Investigation, Methodology, Project administration, Resources, Software, Validation, Visualization, Writing – original draft, Writing – review and editing; Ievgeniia Zagoriy, Bernhard Hampoelz, Janina Baumbach, Investigation, Resources, Writing – review and editing; Xiaojie Zhang, Methodology, Writing – review and editing; Philipp S Erdmann, Methodology, Resources, Writing – review and editing; Christoph W Müller, Funding acquisition, Project administration, Supervision, Writing – review and editing; Martin Beck, Project administration, Supervision; Jürgen M Plitzko, Julia Mahamid, Conceptualization, Funding acquisition, Investigation, Methodology, Project administration, Resources, Supervision, Visualization, Writing – original draft, Writing – review and editing

## Author ORCIDs

Sven Klumpe  http://orcid.org/0000-0002-8350-6503
Herman KH Fung  http://orcid.org/0000-0001-9568-1782
Sara K Goetz  http://orcid.org/0000-0002-9903-3667
Christoph W Müller  http://orcid.org/0000-0003-2176-8337
Martin Beck  http://orcid.org/0000-0002-7397-1321
Jürgen M Plitzko  http://orcid.org/0000-0002-6402-8315
Julia Mahamid  http://orcid.org/0000-0001-6968-041X

## Decision letter and Author response

Decision letter https://doi.org/10.7554/eLife.70506.sa1
Author response https://doi.org/10.7554/eLife.70506.sa2

# Additional files

## Supplementary files

• Supplementary file 1. Selected parameters for milling of micro-expansion joints, lamella rough, and fine milling of five different cell types.

• Supplementary file 2. Correlated lipid droplet positions post-milling and best-fitting fluorescence plane position in the y-direction of the focused ion beam (FIB) image relative to the observed lamella position.

• Transparent reporting form

## Data availability

All code developed in this work is available on GitHub. SerialFIB and a written tutorial can be obtained on: (https://github.com/sklumpe/SerialFIB; copy archived at swh:1:rev:0eaaaf66afa2d803440cea18af-85c444df10478f). A comprehensive video tutorial session is uploaded to YouTube: https://www.youtube.com/watch?v=QR7ngJ0apBk. The documentation is available on GitHub (https://github.com/sklumpe/SerialFIB/blob/main/documentation/SFIB.pdf). A Python script for post-processing of cryo-FIB-SEM volume imaging data as described in the method section is available in the SerialFIB GitHub repository. The Python 3-ported 3DCT is available on: https://github.com/hermankhfung/3dct. New 3DCT functions described in this work, scripts for cryo-FLM virtual slice series creation, 3D point-based registration and transformation of cryo-FLM data with respect to cryo-FIB-SEM data, and bUnwarpJ-based analysis of FIB images before and after milling are available on: https://github.com/hermankhfung/tools3dct. All tomograms and subtomogram averages depicted in the figures are deposited to EMDB under accession codes EMD-13832, EMD-13833, EMD-13834, EMD-13835, EMD-13836, EMD-13837, EMD-13838, EMD-13878, EMD-13879. FIB-SEM volume imaging data for the Sum159 cells and *Chlamydomonas reinhardtii* are deposited in EMPIAR under accession codes

EMPIAR-10847, EMPIAR-10870. FIB, SEM, TEM and fluorescence images used for 3D correlation are available on the BioImage Archive (S-BSST730, S-BSST729). Numerical data for lamella widths and thicknesses presented in Figure 2-figure supplement 4 are tabulated in Figure 2-figure supplement 4-Source Data 1 and 2.

The following dataset was generated:

| Author(s) | Year | Dataset title | Dataset URL | Database and Identifier |
|---|---|---|---|---|
| Klumpe S, Fung HKH, Goetz SK, Plitzko JM, Mahamid J | 2022 | Subtomogram average of 80S ribosomes from a cryo-FIB-lamella of Sum159 human cell line | https://www.ebi.ac.uk/emdb/EMD-13832 | Electron Microscopy Data Bank, EMD-13832 |
| Klumpe S, Fung HKH, Goetz SK, Plitzko JM, Mahamid J | 2022 | Cryo-electron tomograms from cryo-FIB-lamellae of Sum159 human cell line | https://www.ebi.ac.uk/emdb/EMD-13833 | Electron Microscopy Data Bank, EMD-13833 |
| Klumpe S, Fung HKH, Goetz SK, Plitzko JM, Mahamid J | 2022 | Subtomogram average of 80S ribosomes from a cryo-FIB-lamella of Sum159 human cell line prepared after cryo-FIB-SEM volume imaging | https://www.ebi.ac.uk/emdb/EMD-13834 | Electron Microscopy Data Bank, EMD-13834 |
| Klumpe S, Fung HKH, Goetz SK, Plitzko JM, Mahamid J | 2022 | Cryo-electron tomograms from cryo-FIB-lamellae of Sum159 human cell line prepared after cryo-FIB-SEM volume imaging | https://www.ebi.ac.uk/emdb/EMD-13845 | Electron Microscopy Data Bank, EMD-13835 |
| Klumpe S, Fung HKH, Goetz SK, Plitzko JM, Mahamid J | 2022 | Cryo-electron tomogram of a cryo-FIB lamella of a HeLa cell | https://www.ebi.ac.uk/emdb/EMD-13836 | Electron Microscopy Data Bank, EMD-13836 |
| Klumpe S, Fung HKH, Goetz SK, Plitzko JM, Mahamid J | 2022 | Cryo-electron tomogram of a cryo-FIB lamella of Emiliania huxleyi cells | https://www.ebi.ac.uk/emdb/EMD-13837 | Electron Microscopy Data Bank, EMD-13837 |
| Klumpe S, Fung HKH, Goetz SK, Plitzko JM, Mahamid J | 2022 | Cryo-electron tomogram of a 3D correlated lipid droplet in a cryo-FIB-milled HeLa cell | https://www.ebi.ac.uk/emdb/EMD-13838 | Electron Microscopy Data Bank, EMD-13838 |
| Klumpe S, Plitzko JM, Mahamid J | 2022 | Cryo-electron tomogram from a cryo-FIB lift-out lamella from Drosophilia melanogaster egg chamber | https://www.ebi.ac.uk/emdb/EMD-13878 | Electron Microscopy Data Bank, EMD-13878 |
| Klumpe S, Fung HKH, Goetz SK, Plitzko JM, Mahamid J | 2022 | Subtomogram average of D. melanogaster ribosomes from tomograms collected on a cryo-lift-out lamella | https://www.ebi.ac.uk/emdb/EMD-13879 | Electron Microscopy Data Bank, EMD-13879 |
| Klumpe S, Fung HKH, Goetz SK, Plitzko JM, Mahamid J | 2022 | Cryo-FIB-SEM volume in a Sum159 human cell line | https://dx.doi.org/10.6019/EMPIAR-10847 | Electron Microscopy Public Image Archive, 10.6019/EMPIAR-10847 |
| Klumpe S, Fung HKH, Goetz SK, Plitzko JM, Mahamid J | 2022 | Cryo-FIB-SEM data on Chlamydomonas reinhardtii cells | https://dx.doi.org/10.6019/EMPIAR-10870 | Electron Microscopy Public Image Archive, 10.6019/EMPIAR-10870 |
| Klumpe S, Fung HKH, Goetz SK, Plitzko JM, Mahamid J | 2022 | Cryo-correlative fluorescence and FIB-SEM volume imaging of HeLa cells | https://www.ebi.ac.uk/biostudies/studies/S-BSST729 | BioImage Acrhive, S-BSST729 |
| Klumpe S, Fung HKH, Goetz SK, Plitzko JM, Mahamid J | 2022 | Cryogenic 3D-CLEM targeted lamella preparation of HeLa cells | https://www.ebi.ac.uk/biostudies/studies/S-BSST730 | BioImage Acrhive, S-BSST730 |

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

## Appendix 1

## Extending SerialFIB to different microscopes

SerialFIB is open source and uses a Python-based driver to communicate with the FIB-SEM microscope. The driver calls functions directly from the microscope's API. Therefore, for every API, a new driver is needed. As long as the manufacturer provides an API that allows sufficient control of the microscope, the driver developed here for Thermo Fisher instruments can be adapted to use the new API. Functions that need to be adapted are listed in the documentation on GitHub. Alternatively, developers can adapt each function of the *fibsem* class to the new API in the driver's source code.

Our developed driver communicates with Thermo Fisher Scientific instruments via AutoScript 4. SerialFIB was benchmarked on Aquilos and Scios systems that were available to us for the development. Other Thermo Fisher Scientific instruments support AutoScript 4. From the AutoScript technical note, this includes Versa 3D, Quanta FEG, Nova NanoSEM, Teneo, Scios 1, Scios 2, Helios PFIB, Helios G4 PFIB, Helios NG, G4, G5, Apreo, Quattro, and Prisma E. For applications at cryogenic temperature, only the Aquilos, Scios, Quanta, and Helios FIBs are currently used.

When extending SerialFIB to other microscope providers, a driver needs to be developed for the respective API. For Zeiss instruments, the available API is called the Zeiss Remote API. Zeiss Remote API allows development in the Microsoft.NET framework. The programming languages accessible are Visual Basic, Visual C++, or Visual C# (Technology Note 'Zeiss Remote API: Customizing Tool Functionality Using the Remote Application Programming Interface,' S. Chalal & Heiko Stegman, August 2013). Since SerialFIB is Python-based, a wrapper to make the calls to the microscope, for example, in C++, and parsing of the response to Python is required. C/C ++ in Python wrappers are commonly developed (e.g., the framework for GUIs used here, PyQt5 is wrapping Qt), rendering the development of the driver feasible. It would however require significant additional development. For TESCAN instruments, an API called SharkSEM is available and is Python based. Development of a driver would be possible by only changing the syntax within the existing driver for Thermo Fisher Scientific instruments. This is similar for Hitachi instruments, where a Python-based API called EM-Macro is available. JEOL instruments allow for scripting via Python as well. At this point, we are not aware of additional cryo-FIB-SEM instrument providers.

