## [Editor Report]

Since its initial inception, as a sample thinning technique for cryo-electron tomography, cryo-focused ion beam (FIB) milling has developed to include a range of different methodologies. At the moment, there is no dedicated software that is able to integrate all these methodologies and their supporting software packages. Klumpe et al. aimed to alleviate this problem by developing an open-source software tool, SerialFIB. SerialFIB allows users to set up automated protocols for on-grid lamella preparation, FIB-SEM volume imaging, and lift-out trench milling. This work has significant importance for the field as it decouples the need for proprietary software for the execution of highly specialized milling protocols.

---

## [Decision Letter]

**Decision letter after peer review:**

Thank you for submitting your article "A Modular Platform for Streamlining Automated Cryo-FIB Workflows" for consideration by *eLife*. Your article has been reviewed by 3 peer reviewers, and the evaluation has been overseen by a Reviewing Editor and Richard Aldrich as the Senior Editor.

The reviewers have discussed their reviews with one another, and the Reviewing Editor has drafted this to help you prepare a revised submission in the "Tools and Resources" category.

Essential revisions:

The paper's focus, the software package SerialFIB, is very poorly described. All the information on how the software works (lamella/pattern/volume design and script editor) takes up only a small section of the results. Prospective users of the software would like to know how SerialFIB features are used and applied. We acknowledge that the authors have provided videos and supplementary figures in an attempt to introduce the software's features. Unfortunately, these videos are not helpful to give the required insight. More comprehensive videos are needed, which show the automated milling and volume imaging of a sample from the design stage to the final lamella. Informative captions throughout the video should be included so that the procedure is easy to follow. Video S1 intends to demonstrate the workflow, however, it is hard to follow due to little number of explanatory legends. A narration or comment of the different steps would help as well. Since the scripting module and its compatibility with Python is another potential benefit of this software, we think it is important to better showcase its use for the reader.

The authors show that the tool runs on Thermo Fisher instrumentation. For SerialFIB to become widely applicable and to realize its full potential, it would be great if the implementation on instruments of other vendors could be shown – at least a more clear outline of the steps to be taken should be provided.

The insights into the deformation of the sample during the milling were very intriguing. Unfortunately, from the amount of data that is shown, there is too little evidence to make a definitive statements about the importance of grid rigidity for successful targeting ('Our benchmarking results illustrate the importance of having a mechanically stable specimen when performing 3D-CLEM-guided milling'). In particular, there are no replicates of the gold grid results to verify that this outcome is not an outlier. In addition, for us it would be important to have more than two grid types and one sample type before making a statement about the importance of a mechanically stable specimen.

The tools presented in this manuscript are an important contribution to the growing in situ structural biology community. It is great to see the diversity of applications supported by SerialFIB. Despite without doubt representing an immense advance, the authors may want to be more careful not to give the impression that everything is easy. This is especially important since many new labs that are now diving into cryo-ET have previous experience in SPA cryo-EM which is streamlined to a more advanced level, and they may interpret the availability of this platform as a similar level of facilitation.

Please point out the shortcomings of the cryo-CLEM procedure, and that it not suitable for targeted cryo-FIB milling to locate rare and small protein complexes. It's important to more clearly discuss that the way cryo-FM-guided FIB milling currently works is that the positioning of the lamellae, i.e. the milling process, is only roughly targeted, while a more precise correlation between the final lamella and the cryo-FM best fitting z-plane is generated post-milling. It remains a bit unclear how the best fitting z-plane is determined. Line 712 implies that it is empirically chosen, based on visual similarity between image features. This is a potential source of error and requires significant user intuition, which should be explicitly discussed. Moreover, I understand that the targeting is relatively good when stable support grids are used and when relatively large fluorescent objects, such as lipid droplets (estimated 300-500 nm in diameter), are targeted. However, the size of the object to locate (when it is not diffraction-limited) will play a role in how frequently the targeting is successful: this issue should at least be discussed. Furthermore, how is the precision of lamellae positioning (axial correlation precision) affected by the (limited, possibly suboptimal) distribution of the microbeads in z, and the limited axial resolution of the cryo-confocal microscope? I appreciate the discussion of the axial compression of the imaged volume (line 585) being unproblematic for affine transforms on the level of single cells, but I am not convinced that the precision of localisation in z is unaffected, and I think this should be better discussed.

The presented correlation workflow relies on the acquisition of 3D confocal stacks on a cryo-confocal microscope. This is at present not a standard instrument and most labs that use cryo-fluorescence microscopy have wide-field instruments. It would be very helpful if the authors could also provide an example how 2D wide-field fluorescence images can be used during the milling procedure.

The ribosome subtomogram averaged maps are a useful measure for the quality of the data, but the way the maps are presented it is difficult to judge whether they really represent ribosomes, and how good the maps are. At least the FSC curves used to assess the resolutions should be shown in a supplement figure.

*Reviewer #1:*

Since its initial inception, as a sample thinning technique for cryo-electron tomography, cryo-focused ion beam (FIB) milling has developed to include a range of different methodologies. At the moment there is no dedicated software which is able to integrate all these methodologies and their supporting software packages. Klumpe et al. aimed to alleviate this problem by developing an open-source software tool, SerialFIB. SerialFIB allows users to set up automated protocols for on-grid lamella preparation, FIB-SEM volume imaging and lift-out trench milling. In addition, SerialFIB takes advantage of external software packages, such as 3DCT for the correlation of fluorescent light microscopy data and provides a scripting module for users to design their own milling methodologies. The authors shortly describe the newly developed software, which is followed by an extensive showcasing of data produced by using this software.

This paper has significant importance for the field, as it decouples the need for proprietary software for the execution of highly specialized milling protocols. Due to continuous development in the cryo-FIB milling community it will become of importance to have a software that is able to integrate all the new developments into one tool. SerialFIB shows that it is able to automate milling procedures that are commonly used in the FIB milling community. In addition, by providing an example of how the software can interface with a previously existing 3-dimensional correlative light and electron microscopy tool, 3D-CLEM, SerialFIB shows that it may become the 'one-stop shop' for cryo-FIB milling community (similar to SerialEM for tomography data acquisition).

That said, there are a number of major concerns:

1. The paper's focus, the software package SerialFIB, is very poorly described. All the information on how the software works (lamella/pattern/volume design and script editor) takes up only a small section of the results. Prospective users of the software would like to know how SerialFIB features are used and applied. We acknowledge that the authors have provided videos and supplementary figures in an attempt to introduce the software's features. Unfortunately, these videos are not helpful to give the required insight. More comprehensive videos are needed, which show the automated milling and volume imaging of a sample from the design stage to the final lamella. Informative captions throughout the video should be included so that the procedure is easy to follow. Video S1 intends to demonstrate the workflow, however, it is hard to follow due to little number of explanatory legends. A narration or comment of the different steps would help as well. Since the scripting module and its compatibility with Python is another potential benefit of this software, we think it is important to better showcase its use for the reader.

2. While the paper provides a potentially great tool, it does not report conceptual advance regarding methodology or biology. It should therefore be considered as a resource article.

3. The authors show that the tool runs on Thermo Fisher instrumentation. For SerialFIB to become widely applicable and to realize its full potential, it would be great if the implementation on instruments of other vendors could be shown – at least a more clear outline of the steps to be taken should be provided.

4. The insights into the deformation of the sample during the milling were very intriguing. Unfortunately, from the amount of data that is shown, there is too little evidence to make a definitive statements about the importance of grid rigidity for successful targeting ('Our benchmarking results illustrate the importance of having a mechanically stable specimen when performing 3D-CLEM-guided milling'). In particular, there are no replicates of the gold grid results to verify that this outcome is not an outlier. In addition, for us it would be important to have more than two grid types and one sample type before making a statement about the importance of a mechanically stable specimen.

*Reviewer #2:*

This work introduces serialFIB, an open-source tool for the automation of in situ cryo-ET sample preparation using focused ion-beam milling. SerialFIB is comprised of a graphical user interface, a scripting module, and a driver that interprets user-defined actions for the microscope. The driver works with the TFS dual beam instruments. Based on the desired milling protocol, a user can work with either the GUI or the scripting module. The authors provide substantial evidence on how this tool can be programmed to carry out a wide range of existing sample preparation techniques including: lamella preparation on single cells, fluorescence guided lamella targeting, trench milling for cryo-lift out procedures, and cryo-FIB-SEM volume imaging. All the workflows are fully benchmarked from sample preparation all the way to sub-tomogram averaging. The authors have also tested these workflows on multiple different cell types and provided their recipes for the thinning of each specific cell type. The data provided supports the claim that serialFIB makes the execution of these techniques more streamlined and flexible. As an example, the authors take advantage of the streamlined CLEM lamella preparation and study the deformation of the grid support because of exposure to the ion beam and quantify the resulting offsets in the positioning of the lamella. The authors then provide a workflow that will help with the correct registration of FLM volume data with the tomograms of the offset lamella acquired with TEM. This finding alone is of great interest to the researches in the field. In another benchmarking effort, the authors characterize lamella preparation directly following cryo-FIB-SEM volume imaging. They provide a recipe where volume imaging can be used to identify sub-cellular organelles after which a lamella is prepared and imaged in TEM. A reconstruction of the ribosomes in that lamella shows that the quality is comparable to that of a conventionally prepared lamella. They conclude that the radiation damage during volume imaging can be managed and minimized in this workflow.

This work uses previously established techniques to demonstrate the abilities of serialFIB. Being able to carry out such a wide range of techniques proves that serialFIB is flexible and programmable. Furthermore, the impact of serialFIB on the field is established by applying it to important fundamental questions such as sample deformation and radiation damage. This is done in a quantitative manner which reflects the high throughput that is enabled by serialFIB.

Aside from introducing multiple useful tools, many useful recipes are provided in this paper. These include milling parameters for different cell types, milling conditions for volume imaging, and techniques for fluorescence labeling of the cells and the subsequent registration of FLM data with TEM data. This will be of great value to the researchers in the field.

Each workflow is fully tested from milling all the way to sub-tomogram averaging. The authors use ribosomes as standard particles with which they determine the quality of the tomograms. They report the number of sub-tomograms used in each reconstruction and the resolution.

Two important milling side-effects are characterized and quantified in this paper; one is sample deformation and the other is radiation damage. TEM imaging of sparse biological phenomena in cells requires guided lamella preparation with the help of fluorescence tags. The authors show that despite the guided lamella preparation, the final lamella can be offset to the extent that it does not encompass the region of interest anymore. This is traced back to the deformations in the support film that can happen at the area exposed to the ion beam and even in the surrounding areas. Using two different grid types that differ in the continuity of the support film and the element of the grid bars, the authors demonstrate substantial variability in the extent of sample deformation. This is an important lesson for the researchers in the field and it provides a workflow to characterize such deformations for different grid types.

Although volume imaging of chemically fixed specimen is well characterized, cryo-FIB-SEM volume imaging remains challenging both in implementation and in minimizing radiation damage. The authors present an interesting case where a lamella is prepared from a cell that has also gone through volume imaging. By checking the lamella quality with a 3D reconstruction of a specific ribosome, the authors successfully demonstrate a workflow where radiation is optimized to produce enough contrast in the SEM volume stack and also confined such that a lamella can be prepared for TEM only 100 nm below the imaged volume.

*Reviewer #3:*

Cryo-FIB milling for generating thin lamellae amenable to cryo-ET is considered tedious, time-consuming and requiring expert knowledge. Therefore, approaches to automate the process and to interface it better with other imaging modalities such as SEM imaging or cryo-fluorescence imaging are eagerly awaited by the community. This paper and the presented open-source software suite will without doubt contribute to popularising and easing different cryo-FIB SEM applications. Particularly the flexible protocols with tested, recipe-like default parameters will be helpful. Despite these advances, it should be noted that some of the procedures addressed here, in particular cryo-CLEM and cryo-lift-out from high-pressure frozen tissue samples, remain difficult non-routine applications that will still require expert know-how and empirical developments of protocols. This is not to diminish the contributions of this paper but to warn readers that these methods are still neither easy nor fully automated. As the authors note, the software is a foundation for future developments. However several bottlenecks remain unsolved.

In particular, the presented cryo-CLEM procedures are useful for a rough targeting of organelles, but for precisely localising rare or small protein assemblies in cells, the accuracy of the fluorescence-based milling is not sufficient. Currently, this problem is circumvented by empirically finding the best-matching fluorescence z-plane after the lamella has been milled. While this works for relatively abundant and large (non-diffraction limited), easily identifiable structures such as lipid droplets, rare and small structures will more often be missed.

---

## [Author Response]

Essential revisions:The paper's focus, the software package SerialFIB, is very poorly described. All the information on how the software works (lamella/pattern/volume design and script editor) takes up only a small section of the results. Prospective users of the software would like to know how SerialFIB features are used and applied. We acknowledge that the authors have provided videos and supplementary figures in an attempt to introduce the software's features. Unfortunately, these videos are not helpful to give the required insight. More comprehensive videos are needed, which show the automated milling and volume imaging of a sample from the design stage to the final lamella. Informative captions throughout the video should be included so that the procedure is easy to follow. Video S1 intends to demonstrate the workflow, however, it is hard to follow due to little number of explanatory legends. A narration or comment of the different steps would help as well.

We thank the reviewers’ for pointing out the need for a step-by-step guide for the use of SerialFIB. In the manuscript, we aimed to describe the developed software architecture and its usefulness in executing multiple common use-cases. We now include a new figure (Figure 2—figure supplement 1) to illustrate in more detail the procedures performed by SerialFIB to execute the automated milling and volume imaging.

For prospective users of the software, a step-by-step tutorial for setting up all use-cases described in the manuscript is available on GitHub (https://github.com/sklumpe/SerialFIB), where the software is hosted and must be accessed for installation. This tutorial has already been tested and successfully used by other research sites independently. A 50-minutes recorded tutorial session has also been uploaded to Youtube: https://www.youtube.com/watch?v=QR7ngJ0apBk. Detailed documentation on the usage of the Script Editor is also available on GitHub. These can be downloaded at:

https://github.com/sklumpe/SerialFIB/blob/main/20210428_SerialFIB_Tutorial.pdf

https://github.com/sklumpe/SerialFIB/blob/main/documentation/SFIB.pdf

We have updated the Data Availability statement to reflect these tutorials.

The software and the accompanying tutorials (including Video S1) are likely to be subject to change in future versions to implement additional features, and will then be released on GitHub accordingly. We therefore refrain from providing a single tutorial together with the manuscript and removed Video S1.

Since the scripting module and its compatibility with Python is another potential benefit of this software, we think it is important to better showcase its use for the reader.

An example script for the Script Editor is provided with explanations in Section 4 of the documentation on GitHub. Additional scripts are available on GitHub. Figure 1—figure supplement 2B depicts the Script Editor interface. To better showcase the Script Editor as suggested, we now refer to the script collection on GitHub, which has been updated to include a pilot script for automating the Waffle method (1) with accompanying explanations in the documentation which we will continue to develop.

The authors show that the tool runs on Thermo Fisher instrumentation. For SerialFIB to become widely applicable and to realize its full potential, it would be great if the implementation on instruments of other vendors could be shown – at least a more clear outline of the steps to be taken should be provided.

SerialFIB uses a Python-based driver to communicate with microscopes. The Driver calls functions directly from the microscope’s application programming interface (API). Therefore, for every API, a new Driver is needed. As long as a manufacturer provides an API that allows sufficient control of the microscope, the driver developed here for Thermo Fisher instruments can be adapted to use the new API. Functions that need to be adapted are listed in Section 7 of the documentation on GitHub. Alternatively, developers adapt each function of the *fibsem* class to the new API in the driver’s source code. A primary motivation to make our software open-source was to support continued development in this regard. We would be delighted to support interested colleagues in adapting SerialFIB to other instruments. We have included further information in Appendix 1 to explain what is required to implement SerialFIB on other instruments.

The insights into the deformation of the sample during the milling were very intriguing. Unfortunately, from the amount of data that is shown, there is too little evidence to make a definitive statements about the importance of grid rigidity for successful targeting ('Our benchmarking results illustrate the importance of having a mechanically stable specimen when performing 3D-CLEM-guided milling'). In particular, there are no replicates of the gold grid results to verify that this outcome is not an outlier. In addition, for us it would be important to have more than two grid types and one sample type before making a statement about the importance of a mechanically stable specimen.

We thank the reviewers for raising this critical point. As pointed out by the reviewers, 3D-targeted FIB milling and retention of structures of interest in lamellae are far from being routine and easy. Therefore, providing a systematic and quantitative evaluation of multiple grid types, support, and the effect of ice thickness on different cell types would not be feasible within the scope of this work. We believe that even though these results are only qualitative observations, they are still important factors for the successful use of the automation. We have replaced the phrase "benchmarking results” with “observations”, and state that a more systematic evaluation will be needed to confirm our observations.

The tools presented in this manuscript are an important contribution to the growing in situ structural biology community. It is great to see the diversity of applications supported by SerialFIB. Despite without doubt representing an immense advance, the authors may want to be more careful not to give the impression that everything is easy. This is especially important since many new labs that are now diving into cryo-ET have previous experience in SPA cryo-EM which is streamlined to a more advanced level, and they may interpret the availability of this platform as a similar level of facilitation.

This certainly was not our intention and we do not think the paper gives an impression that cryo-FIB milling is easy. While we state that automation can improve the “performance, throughput, and applicability of [current] cryo-FIB methods” (Line 56), we describe the complexity of each application and critical considerations for various aspects of the sample preparation beyond the automation itself (Lines 597–646).

Please point out the shortcomings of the cryo-CLEM procedure, and that it not suitable for targeted cryo-FIB milling to locate rare and small protein complexes. It's important to more clearly discuss that the way cryo-FM-guided FIB milling currently works is that the positioning of the lamellae, i.e. the milling process, is only roughly targeted, while a more precise correlation between the final lamella and the cryo-FM best fitting z-plane is generated post-milling.

We now emphasise in Line 273 that the targeting of structures by cryo-FLM-guided FIB milling remains technically challenging and that the approaches presented only aim to improve “throughput and usability”. We also state explicitly in Line 359 that refinement of the correlation based on lipid droplet staining is performed “post-milling”. The statement concerning targeting of “small and rare structures” in the Discussion has been removed.

It remains a bit unclear how the best fitting z-plane is determined. Line 712 implies that it is empirically chosen, based on visual similarity between image features. This is a potential source of error and requires significant user intuition, which should be explicitly discussed.

We thank the reviewers for pointing this out. We have revised the text and now emphasise that the best fitting plane is determined “qualitatively” (Line 356). To illustrate the process of determining the best fitting plane, we have added new panels F–H to Figure 3 supplement 2 showing a TEM lamella image overlaid with projections of virtual slices corresponding to different heights in the FIB image.

Moreover, I understand that the targeting is relatively good when stable support grids are used and when relatively large fluorescent objects, such as lipid droplets (estimated 300-500 nm in diameter), are targeted. However, the size of the object to locate (when it is not diffraction-limited) will play a role in how frequently the targeting is successful: this issue should at least be discussed.

We welcome this comment from the reviewer. Descriptions of registration errors have been formalised and extended by many in the field of medical imaging, with particular focus on image-guidance systems for surgery (2). Like in surgery, the likelihood of successful targeting by FLM-guided FIB milling will depend on the target registration error, defined as the deviation of a target point in the registered image from its true position (2, 3). This error depends on the target position, the placement of fiducials and the error with which fiducials are localized (more below). Fiducials need to be big enough to spot in FIB and SEM images when embedded in ice, whereas they also need to be bright enough for accurate Gaussian-based fitting of their centres in the fluorescence modality (3). We have thus chosen micron-sized beads for the correlative workflow. As for the size of the target object, we expect this to affect targeting in at least two ways: (i) when the size is below or close to the registration error, the likelihood of success in targeting will drop; (ii) if the fluorescence signal is weak because the target object is small or contains few fluorophores, there will be a greater error in the localisation of the target itself. We now elaborate on these concepts in the Discussion (Line 621–639). Finally, the size of most lipid droplets targeted in this study does not exceed the theoretical resolution limit of the cryo-FLM system used (up to 442 nm laterally and 1.9 μm axially, now stated in Line 267–269, and similar to the values previously reported in (3)). Thus, we feel that lipid droplets represent a good case study for testing the presented workflow (Line 343-345). An experimental measurement of the point spread function (PSF) on the system used based on 200-nm Tetraspeck beads is now also provided in Line 752.

Furthermore, how is the precision of lamellae positioning (axial correlation precision) affected by the (limited, possibly suboptimal) distribution of the microbeads in z, and the limited axial resolution of the cryo-confocal microscope?

It has been shown for rigid-body registration that the target registration error increases as the target moves away from the principal axes of the fiducial distribution (2). If the distribution of fiducials is anisotropic, for example, limited in z, then we would expect the error to increase. Target registration error also scales with fiducial localisation error, which is highly dependent on the point spread function. Axial resolution in cryo-FLM is commonly worse than lateral resolution (3), and these values are now provided in the revised manuscript. Therefore, when imaging side-on with the FIB with respect to the fluorescence xy-plane, we would expect the highest registration errors in the 3D-to-2D correlation.

Potier et al. (4) describe a multivariate linear regression-based approach to estimate registration errors for any target point in an affine registration. The approach also considers anisotropic fiducial localisation errors, e.g., asymmetric PSFs. We anticipate that this will provide a suitable framework for analysing the 3D-to-2D correlation performed here. However, adaptation will be required to take into account the dimensionality reduction that is part of the registration process. Thus, a systematic study of the effects of bead distribution and imaging resolution on registration error is beyond the scope of this work.

I appreciate the discussion of the axial compression of the imaged volume (line 585) being unproblematic for affine transforms on the level of single cells, but I am not convinced that the precision of localisation in z is unaffected, and I think this should be better discussed.

As stated above and now more explicitly in the Discussion (Line 631–639), the PSF will contribute to the fiducial localisation error. As demonstrated for super-resolution fluorescence imaging, the PSF becomes more asymmetric at increasing imaging depth and can be empirically corrected (5). We have not implemented such corrections in the current workflow. Thus, fiducial localisation error due to poor axial resolution and depth-induced aberrations can be expected. However, in the volume registered, we did not identify a significant correlation to indicate an increase in residual error with fluorescence imaging depth (Author response image 1), leading to our statement regarding the affine transforms in single cells. To provide a more complete picture, we now describe in the Discussion (Line 640–645) additional errors that can affect the registration performed: non-uniform slice thickness in the FIB-SEM volume due to curtaining, fiducial localisation error due to the 100-nm FIB milling steps and the unweighted centroid-based method for determining centres.

**Author response image 1. sa2fig1:** Absolute residuals from the point-based affine 3D registration between FIB-SEM and fluorescence volumes of HeLa cells presented in Figure 4—figure supplement 2, plotted against their z-position in the fluorescence volume for each point. Correlation between residual error and imaging depth is weak in this instance, as reflected in the squared Pearson correlation coefficient.

The presented correlation workflow relies on the acquisition of 3D confocal stacks on a cryo-confocal microscope. This is at present not a standard instrument and most labs that use cryo-fluorescence microscopy have wide-field instruments. It would be very helpful if the authors could also provide an example how 2D wide-field fluorescence images can be used during the milling procedure.

The presented workflow can be used with any type of fluorescence images, regardless of whether they are single 2D images or z-stacks, and whether they were acquired on a widefield or confocal system. Figure 5 demonstrates how 2D fluorescence images can be used for correlation in 3DCT and how the ensuing output can be used to define milling sites in SerialFIB. To ensure good correlation accuracy when the angle between fluorescence imaging and FIB milling differs greatly, such as in the on-grid milling and volume imaging cases that we presented, it is important to acquire z-stacks to enable 3D correlation. While in confocal imaging the use of a focused laser and pinhole helps to remove out-of-plane fluorescence, widefield images can benefit from deconvolution to achieve almost comparable resolutions (6). Thus, on a widefield system, we would still recommend z-stack acquisition followed by deconvolution to help improve accuracy and precision, respectively, in 3D targeting.

The ribosome subtomogram averaged maps are a useful measure for the quality of the data, but the way the maps are presented it is difficult to judge whether they really represent ribosomes, and how good the maps are. At least the FSC curves used to assess the resolutions should be shown in a supplement figure.

FSC curves for all subtomogram averages presented are now available in Figure 6. All averages and tomograms shown are deposited on EMDB with accession codes (EMD-13832, EMD-13833, EMD-13836, EMD-13837, EMD-13838, EMD-13834, EMD-13835, EMD-13878, EMD-13879) and on EMPIAR (EMPIAR-10847, EMPIAR-10870).

Reviewer #1:Since its initial inception, as a sample thinning technique for cryo-electron tomography, cryo-focused ion beam (FIB) milling has developed to include a range of different methodologies. At the moment there is no dedicated software which is able to integrate all these methodologies and their supporting software packages. Klumpe et al. aimed to alleviate this problem by developing an open-source software tool, SerialFIB. SerialFIB allows users to set up automated protocols for on-grid lamella preparation, FIB-SEM volume imaging and lift-out trench milling. In addition, SerialFIB takes advantage of external software packages, such as 3DCT for the correlation of fluorescent light microscopy data and provides a scripting module for users to design their own milling methodologies. The authors shortly describe the newly developed software, which is followed by an extensive showcasing of data produced by using this software.This paper has significant importance for the field, as it decouples the need for proprietary software for the execution of highly specialized milling protocols. Due to continuous development in the cryo-FIB milling community it will become of importance to have a software that is able to integrate all the new developments into one tool. SerialFIB shows that it is able to automate milling procedures that are commonly used in the FIB milling community. In addition, by providing an example of how the software can interface with a previously existing 3-dimensional correlative light and electron microscopy tool, 3D-CLEM, SerialFIB shows that it may become the 'one-stop shop' for cryo-FIB milling community (similar to SerialEM for tomography data acquisition).That said, there are a number of major concerns:1. The paper's focus, the software package SerialFIB, is very poorly described. All the information on how the software works (lamella/pattern/volume design and script editor) takes up only a small section of the results. Prospective users of the software would like to know how SerialFIB features are used and applied. We acknowledge that the authors have provided videos and supplementary figures in an attempt to introduce the software's features. Unfortunately, these videos are not helpful to give the required insight. More comprehensive videos are needed, which show the automated milling and volume imaging of a sample from the design stage to the final lamella. Informative captions throughout the video should be included so that the procedure is easy to follow. Video S1 intends to demonstrate the workflow, however, it is hard to follow due to little number of explanatory legends. A narration or comment of the different steps would help as well. Since the scripting module and its compatibility with Python is another potential benefit of this software, we think it is important to better showcase its use for the reader.

We thank the reviewers’ for pointing out the need for a step-by-step guide for the use of SerialFIB. In the manuscript, we aimed to describe the developed software architecture and its usefulness in executing multiple common use-cases. We now include a new figure (Figure 2—figure supplement 1) to illustrate in more detail in a flowchart the procedures performed by SerialFIB to execute the automated milling and volume imaging.

For prospective users of the software, a step-by-step tutorial for setting up all use-cases described in the manuscript is available on GitHub (https://github.com/sklumpe/SerialFIB), where the software is hosted and must be accessed for installation. This tutorial has already been tested and successfully used by other research sites independently. A 50-minutes recorded tutorial session has also been uploaded to Youtube: https://www.youtube.com/watch?v=QR7ngJ0apBk. Detailed documentation on the usage of the Script Editor is also available on GitHub. These can be downloaded at: https://github.com/sklumpe/SerialFIB/blob/main/20210428_SerialFIB_Tutorial.pdf

https://github.com/sklumpe/SerialFIB/blob/main/documentation/SFIB.pdf

We have updated the Data Availability statement to reflect these tutorials.

An example script for the Script Editor is provided with explanations in Section 4 of the GitHub documentation. Functions that can be called from the Script Editor are listed in Section 7 of the documentation: https://github.com/sklumpe/SerialFIB/blob/main/documentation/SFIB.pdf

A point-by-point explanation of a pilot script for the Waffle milling method is provided in the documentation mentioned above and given in the scripting examples: https://github.com/sklumpe/SerialFIB/tree/main/ScriptingExamples

The software and the accompanying tutorials are likely to be subject to change in future versions to implement additional features, and will be released on GitHub.

2. While the paper provides a potentially great tool, it does not report conceptual advance regarding methodology or biology. It should therefore be considered as a resource article.

We thank the reviewer for acknowledging the potential of our development to serve as a useful tool for the community and are happy to have the manuscript considered as a resource article.

3. The authors show that the tool runs on Thermo Fisher instrumentation. For SerialFIB to become widely applicable and to realize its full potential, it would be great if the implementation on instruments of other vendors could be shown – at least a more clear outline of the steps to be taken should be provided.

SerialFIB uses a Python-based driver to communicate with microscopes. The Driver calls functions directly from the microscope’s application programming interface (API). Therefore, for every API, a new Driver is needed. As long as a manufacturer provides an API that allows sufficient control of the microscope, the driver developed here for Thermo Fisher instruments can be adapted to use the new API. Functions that need to be adapted are listed in Section 6 of the documentation on GitHub. Alternatively, developers adapt each function of the *fibsem* class to the new API in the driver’s source code. A primary motivation to make our software open-source was to support continued development in this regard. We would be delighted to support interested colleagues in adapting SerialFIB to other instruments. We have added Appendix 1 to explain what is required to implement SerialFIB on other instruments.

4. The insights into the deformation of the sample during the milling were very intriguing. Unfortunately, from the amount of data that is shown, there is too little evidence to make a definitive statements about the importance of grid rigidity for successful targeting ('Our benchmarking results illustrate the importance of having a mechanically stable specimen when performing 3D-CLEM-guided milling'). In particular, there are no replicates of the gold grid results to verify that this outcome is not an outlier. In addition, for us it would be important to have more than two grid types and one sample type before making a statement about the importance of a mechanically stable specimen.

We thank the reviewers for raising this critical point. 3D-targeted FIB milling and retention of structures of interest in lamellae are far from being routine and easy. Therefore, providing a systematic and quantitative evaluation of multiple grid types, support, and the effect of ice thickness on different cell types would not be feasible within the scope of this work. We believe that even though these results are only qualitative observations, they are still important factors for the successful use of the automation. We have replaced the phrase "benchmarking results” with “observations”, and state that a more systematic evaluation will be needed to confirm our observations.

Reviewer #2:This work introduces serialFIB, an open-source tool for the automation of in situ cryo-ET sample preparation using focused ion-beam milling. SerialFIB is comprised of a graphical user interface, a scripting module, and a driver that interprets user-defined actions for the microscope. The driver works with the TFS dual beam instruments. Based on the desired milling protocol, a user can work with either the GUI or the scripting module. The authors provide substantial evidence on how this tool can be programmed to carry out a wide range of existing sample preparation techniques including: lamella preparation on single cells, fluorescence guided lamella targeting, trench milling for cryo-lift out procedures, and cryo-FIB-SEM volume imaging. All the workflows are fully benchmarked from sample preparation all the way to sub-tomogram averaging. The authors have also tested these workflows on multiple different cell types and provided their recipes for the thinning of each specific cell type. The data provided supports the claim that serialFIB makes the execution of these techniques more streamlined and flexible. As an example, the authors take advantage of the streamlined CLEM lamella preparation and study the deformation of the grid support because of exposure to the ion beam and quantify the resulting offsets in the positioning of the lamella. The authors then provide a workflow that will help with the correct registration of FLM volume data with the tomograms of the offset lamella acquired with TEM. This finding alone is of great interest to the researches in the field. In another benchmarking effort, the authors characterize lamella preparation directly following cryo-FIB-SEM volume imaging. They provide a recipe where volume imaging can be used to identify sub-cellular organelles after which a lamella is prepared and imaged in TEM. A reconstruction of the ribosomes in that lamella shows that the quality is comparable to that of a conventionally prepared lamella. They conclude that the radiation damage during volume imaging can be managed and minimized in this workflow.This work uses previously established techniques to demonstrate the abilities of serialFIB. Being able to carry out such a wide range of techniques proves that serialFIB is flexible and programmable. Furthermore, the impact of serialFIB on the field is established by applying it to important fundamental questions such as sample deformation and radiation damage. This is done in a quantitative manner which reflects the high throughput that is enabled by serialFIB.Aside from introducing multiple useful tools, many useful recipes are provided in this paper. These include milling parameters for different cell types, milling conditions for volume imaging, and techniques for fluorescence labeling of the cells and the subsequent registration of FLM data with TEM data. This will be of great value to the researchers in the field.Each workflow is fully tested from milling all the way to sub-tomogram averaging. The authors use ribosomes as standard particles with which they determine the quality of the tomograms. They report the number of sub-tomograms used in each reconstruction and the resolution.Two important milling side-effects are characterized and quantified in this paper; one is sample deformation and the other is radiation damage. TEM imaging of sparse biological phenomena in cells requires guided lamella preparation with the help of fluorescence tags. The authors show that despite the guided lamella preparation, the final lamella can be offset to the extent that it does not encompass the region of interest anymore. This is traced back to the deformations in the support film that can happen at the area exposed to the ion beam and even in the surrounding areas. Using two different grid types that differ in the continuity of the support film and the element of the grid bars, the authors demonstrate substantial variability in the extent of sample deformation. This is an important lesson for the researchers in the field and it provides a workflow to characterize such deformations for different grid types.Although volume imaging of chemically fixed specimen is well characterized, cryo-FIB-SEM volume imaging remains challenging both in implementation and in minimizing radiation damage. The authors present an interesting case where a lamella is prepared from a cell that has also gone through volume imaging. By checking the lamella quality with a 3D reconstruction of a specific ribosome, the authors successfully demonstrate a workflow where radiation is optimized to produce enough contrast in the SEM volume stack and also confined such that a lamella can be prepared for TEM only 100 nm below the imaged volume.

We are grateful to the reviewer’s positive evaluation of the new automation capabilities that SerialFIB now provides to the growing in situ structural biology community. We look forward to a combined effort to develop open-source drivers together to make this software applicable to microscopes from other providers, to facilitate the development and implementation of new methods. We strongly believe that it is only through a community effort that systematic testing and evaluation of the preliminary results provided here on local specimen deformation and volume imaging as an additional correlative imaging modality can be achieved, given the current challenging state of these techniques.

Reviewer #3:Cryo-FIB milling for generating thin lamellae amenable to cryo-ET is considered tedious, time-consuming and requiring expert knowledge. Therefore, approaches to automate the process and to interface it better with other imaging modalities such as SEM imaging or cryo-fluorescence imaging are eagerly awaited by the community. This paper and the presented open-source software suite will without doubt contribute to popularising and easing different cryo-FIB SEM applications. Particularly the flexible protocols with tested, recipe-like default parameters will be helpful. Despite these advances, it should be noted that some of the procedures addressed here, in particular cryo-CLEM and cryo-lift-out from high-pressure frozen tissue samples, remain difficult non-routine applications that will still require expert know-how and empirical developments of protocols. This is not to diminish the contributions of this paper but to warn readers that these methods are still neither easy nor fully automated. As the authors note, the software is a foundation for future developments. However several bottlenecks remain unsolved.

We fully acknowledge the enormous challenge of specimen preparations from high-pressure frozen samples, and strongly believe that the development of our automation facilitates in reducing the tedious time-consuming steps of trench preparation. This is a critical step underway to improve the downstream steps towards more streamlined workflows for these challenging samples. For example, the lift-out method has previously been automated in the material sciences. We believe that with a community effort to develop the needed methods, similar capabilities are achievable at cryogenic temperature in the future.

In particular, the presented cryo-CLEM procedures are useful for a rough targeting of organelles, but for precisely localising rare or small protein assemblies in cells, the accuracy of the fluorescence-based milling is not sufficient. Currently, this problem is circumvented by empirically finding the best-matching fluorescence z-plane after the lamella has been milled. While this works for relatively abundant and large (non-diffraction limited), easily identifiable structures such as lipid droplets, rare and small structures will more often be missed.

We thank the reviewer for this critique. In the manuscript’s Discussion, we point to a potential solution whereby “easy to target” intracellular structures, such as lipid droplets in our case, can be used as internal fiducials in multimodal imaging workflows to localise smaller structures of interest. Further development is needed to make the identification of small and rare biological structures possible. Such developments would benefit from super resolution microscopy techniques (12), integrated light microscopy (13) solutions and more stable support (14, 15) and stages.

References:

1. Kelley K, Raczkowski AM, Klykov O, Jaroenlak P, Bobe D, Kopylov M, et al. Waffle Method: A general and flexible approach for FIB-milling small and anisotropically oriented samples. bioRxiv. 2021:2020.10.28.359372.

2. West JB, Fitzpatrick JM. The Distribution of Target Registration Error in Rigid-Body, Point-Based Registration. Information Processing in Medical Imaging. 1999:460–5.

3. Arnold J, Mahamid J, Lucic V, de Marco A, Fernandez JJ, Laugks T, et al. Site-Specific Cryo-focused Ion Beam Sample Preparation Guided by 3D Correlative Microscopy. Biophys J. 2016;110(4):860-9.

4. Potier G, Lavancier F, Kunne S, Paul-Gilloteaux P. A registration error estimation framework for correlative imaging. arXiv preprint arXiv:210306256. 2021.

5. Li Y, Wu Y-L, Hoess P, Mund M, Ries J. Depth-dependent PSF calibration and aberration correction for 3D single-molecule localization. Biomed Opt Express. 2019;10(6):2708-18.

6. Shaw PJ. Comparison of widefield/deconvolution and confocal microscopy for three-dimensional imaging. Handbook of biological confocal microscopy: Springer; 2006. p. 453-67.

7. Schaffer M, Engel BD, Laugks T, Mahamid J, Plitzko JM, Baumeister W. Cryo-focused Ion Beam Sample Preparation for Imaging Vitreous Cells by Cryo-electron Tomography. Bio Protoc. 2015;5(17).

8. Wolff G, Limpens RWAL, Zheng S, Snijder EJ, Agard DA, Koster AJ, et al. Mind the gap: Micro-expansion joints drastically decrease the bending of FIB-milled cryo-lamellae. J Struct Biol. 2019;208(3):107389.

9. Schaffer M, Mahamid J, Engel BD, Laugks T, Baumeister W, Plitzko JM. Optimized cryo-focused ion beam sample preparation aimed at in situ structural studies of membrane proteins. J Struct Biol. 2017;197(2):73-82.

10. Puck TT, Marcus PI, Cieciura SJ. Clonal growth of mammalian cells in vitro growth characteristics of colonies from single HeLa cells with and without a feeder layer. Journal of Experimental Medicine. 1956;103(2):273-84.

11. Weiss D, Schneider G, Niemann B, Guttmann P, Rudolph D, Schmahl G. Computed tomography of cryogenic biological specimens based on X-ray microscopic images. Ultramicroscopy. 2000;84(3-4):185-97.

12. Moser F, Prazak V, Mordhorst V, Andrade DM, Baker LA, Hagen C, et al. Cryo-SOFI enabling low-dose super-resolution correlative light and electron cryo-microscopy (vol 116, pg 4804, 2019). P Natl Acad Sci USA. 2021;118(10).

13. Gorelick S, Buckley G, Gervinskas G, Johnson TK, Handley A, Caggiano MP, et al. PIE-scope, integrated cryo-correlative light and FIB/SEM microscopy. *eLife*. 2019;8:e45919.

14. Russo CJ, Passmore LA. Progress towards an optimal specimen support for electron cryomicroscopy. Curr Opin Struct Biol. 2016;37:81-9.

15. Russo CJ, Passmore LA. Ultrastable gold substrates: Properties of a support for high-resolution electron cryomicroscopy of biological specimens. Journal of Structural Biology. 2016;193(1):33-44.